# Segment Anyword: Mask Prompt Inversion for Open-Set Grounded Segmentation

**Zhihua Liu** [1]  **Amrutha Saseendran** [2]  **Lei Tong** [2]  **Xilin He** [3]  **Fariba Yousefi** [2]  **Nikolay Burlutskiy** [2]  **Dino Oglic** [2]
**Tom Diethe** [2]  **Philip Teare** [2]  **Huiyu Zhou** [1]  **Chen Jin** [2]

## Abstract

Open-set image segmentation poses a significant challenge because existing methods often demand extensive training or fine-tuning and generally struggle to segment unified objects consistently across diverse text reference expressions. Motivated by this, we propose Segment Anyword, a novel training-free visual concept prompt learning approach for open-set language grounded segmentation that relies on token-level cross-attention maps from a frozen diffusion model to produce segmentation surrogates or *mask prompts*, which are then refined into targeted object masks. Initial prompts typically lack coherence and consistency as the complexity of the image-text increases, resulting in suboptimal mask fragments. To tackle this issue, we further introduce a novel linguistic-guided visual prompt regularization that binds and clusters visual prompts based on sentence dependency and syntactic structural information, enabling the extraction of robust, noise-tolerant mask prompts, and significant improvements in segmentation accuracy. The proposed approach is effective, generalizes across different open-set segmentation tasks, and achieves state-of-the-art results of 52.5 (+6.8 relative) mIoU on Pascal Context 59, 67.73 (+25.73 relative) cIoU on gRef-COCO, and 67.4 (+1.1 relative to fine-tuned methods) mIoU on GranDf, which is the most complex open-set grounded segmentation task in the field.

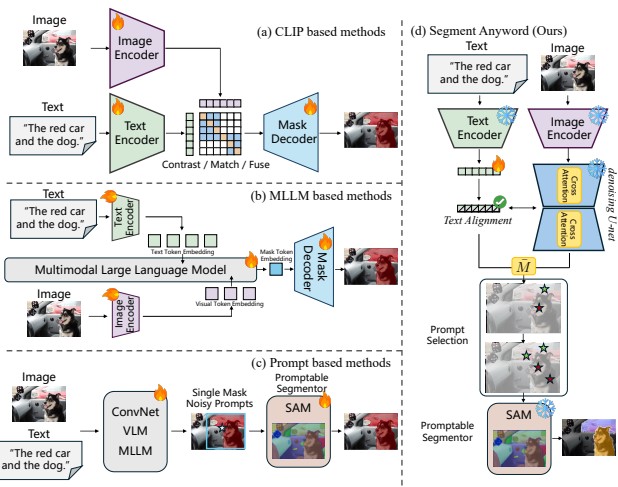

*Figure 1.* Key comparison of multi-modal open-set image segmentation architectures. (a) CLIP based methods (Wang et al., 2022; Xu et al., 2023c) (b) MLLM based methods (Rasheed et al., 2024; Lai et al., 2024; Zhang et al., 2024). (c) Previous prompt learning based methods (Chen et al., 2024; Lin et al., 2024) (d) Our Segment Anyword's architecture, which possesses a simple design for effective arbitrary syntax level grounded segmentation capability with minimal visual prompt optimizing efforts. Project page, code, and data are available at https://zhihualiued.github.io/segment_anyword

## 1. Introduction

Image segmentation plays a critical role in intelligent systems, enabling perceptual applications in various domains such as autonomous driving (Cordts et al., 2016; Geiger et al., 2013), robotic manipulation (Butler et al., 2017; Bruce et al., 2000) and computer-assisted diagnosis (Menze et al., 2014; Litjens et al., 2017). Traditional neural networks are supervised-trained to perform dense pixel-wise classification to delineate the object mask (Long et al., 2015; Ronneberger et al., 2015; He et al., 2017). Although these networks succeed in most cases, they *struggle at handing novel objects in novel domains*, *e.g.* classifying wild animals or new biomarkers, due to a fixed close-set of training data samples and labels (Liu et al., 2020; Wu et al., 2024).

[1]School of Computing and Mathematical Sciences, University of Leicester, UK [2]Centre for AI, Data Science & Artificial Intelligence, BioPharmaceuticals R&D, AstraZeneca, Cambridge, UK [3]Shenzhen University. Correspondence to: Huiyu Zhou <hz143@leicester.ac.uk>, Chen Jin <chen.jin@astrazeneca.com>.

*Proceedings of the 42nd International Conference on Machine Learning*, Vancouver, Canada. PMLR 267, 2025. Copyright 2025 by the author(s).

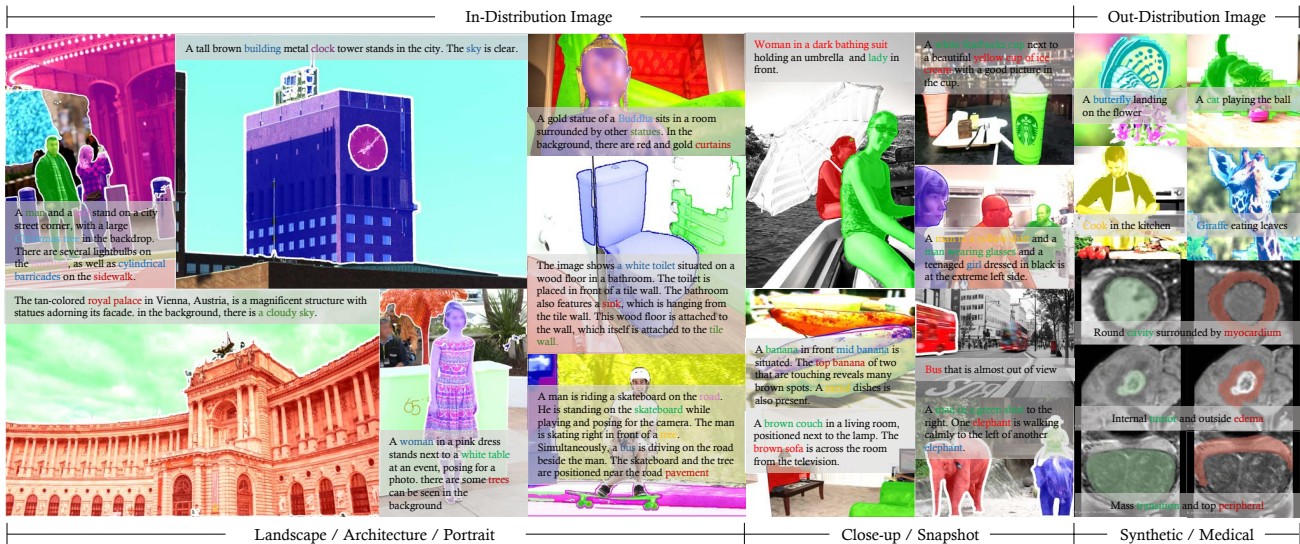

Figure 2. **We propose Segment Anyword for Open-Set Grounded Segmentation.** Segment Anyword is multi-modal promptable image segmentor solely built on the semantic prior knowledge extracted from a frozen diffusion model. Segment Anyword demonstrate superior performance across multiple multi-modal segmentation task, including 1) reference image segmentation (refCOCO, gRefCOCO), 2) complex grounding segmentation (GranDf) and 3) OOD Medical Image Segmentation. Best viewed in colors and zoomed-in.

Recent advancements in vision-language models (VLMs) and multi-modal large language models (MLLMs) have demonstrated a remarkable progress on learning transferable multi-modal representations (Radford et al., 2021; Ho et al., 2020). Related works utilized VLMs (Zhou et al., 2022a; Wang et al., 2022; Xu et al., 2022; 2023c; Sun et al., 2024) or MLLMs (Lai et al., 2024; Rasheed et al., 2024; Xia et al., 2024; Zhang et al., 2024), pre-trained on large-scale image-text datasets, to address open-set segmentation and classify novel objects and categories by training or fine-tuning additional encoders and decoders (Figure 1 (a-c), detailed in Appendix B). Such methods commonly require expensive training or fine-tuning resources and struggle with inferior performance when encountering words or phrases unseen in the training data where diverse terms are used to describe the same object, *e.g.* everyday terms versus domain-specific terminology. We verified this through a motivational study (Figure 3), which highlights the need for a *flexible and adaptive multi-modal segmentation framework that is robust to text variation and minimizes ambiguous segmentation of unified visual concepts.*

Generative models show promise in adapting to unseen objects by treating them as few shot concepts, linking them to learnable textual embeddings and generating customized images (Gal et al., 2022). In this paper, we aim to leverage this strong adaptive capability for image segmentation and propose **Segment Anyword**, which is an efficient framework that takes an off-the-shelf diffusion model for visual concept prompt learning to perform open-set language grounded segmentation (Figure 1 (d)). Segment Anyword capitalizes on

the scalability of the diffusion model, which allows seamless inverse adaptation to new words and visual concepts with minimal effort (Figure 2). By collecting averaged cross-attention maps, which facilitates text-image interaction as a segmentation prior, we randomly sample points from each map as mask prompts for downstream mask generators, such as Segment Anything Model (SAM) (Kirillov et al., 2023). We observe that the initial visual prompts may lack coherence and consistency, resulting inaccurate word-to-mask mapping with suboptimal segmentation fragments. To tackle this, we introduced novel linguistics-based regularization to efficiently bind and cluster mask prompts. The main contributions of this work are summarized as follows:

(1) Our motivational studies revealed that the current VLM and MLLM-based approaches *struggle with diverse terms during open-set segmentation*, resulting in confounded mask outputs, underscores the need for adaptive methods to learn new visual concepts at a low cost.

(2) We propose Segment Anyword, a novel training-free prompt-learning framework for open-set language grounded segmentation performing purely at test-time. Segment Anyword utilizes the cross-attention from a pre-trained diffusion model, which encodes token-level localization of visual concepts, as object prompts to guide downstream segmentation.

(3) To improve word-to-mask mapping accuracy, we propose novel linguistics-guided visual prompt regularization techniques for prompt binding and clustering. This approach allows the integration of linguistic knowledge into visual prompts, enhancing the effectiveness of incorporating text

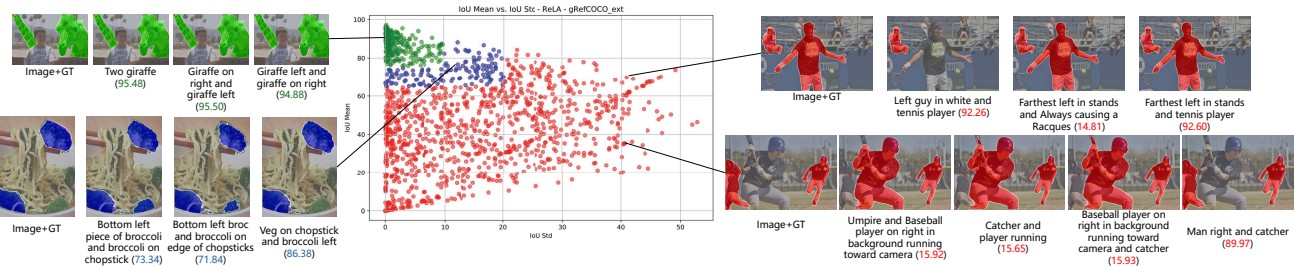

*Figure 3.* **Motivational Study** on the multi-object reference segmentation ReLA (Liu et al., 2023). We categorize data points into easy (green), medium (blue), and hard (red) samples to achieve accurate and stable segmentation, based on the retained IoU mean and standard deviation, which are calculated across the caption dimension. The IoU for each image-text pair, compared to the ground truth, is shown in brackets. Our study validates the presence of segmentation variability, highlighting the challenges in generating accurate and stable segmentation masks with free-form text reference descriptions.

*Table 1.* Comparison of existed multi-modal image segmentation technologies with our proposed method. From left to right: Open-Vocabulary (V), Open-Reference (R), Word-Grounding to index word with object region (G), Training-Free (T), Fine Tuning-Free (F), Localize Novel Concept that has not been encountered during training (C), # Trainable Parameters (#P).

| Method | V | R | G | T | F | C | #P |
|---|---|---|---|---|---|---|---|
| MattNet (Yu et al., 2018) | ✓ | ✓ | ✗ | ✗ | ✗ | ✗ | 27.57M |
| CRIS (Wang et al., 2022) | ✓ | ✓ | ✗ | ✗ | ✗ | ✗ | 40.42M |
| ETRIS (Xu et al., 2023c) | ✓ | ✓ | ✓ | ✓ | ✗ | ✗ | 1.39M |
| GLaMM (Xu et al., 2023c) | ✓ | ✓ | ✓ | ✗ | ✗ | ✗ | ∼130M |
| LISA (Lai et al., 2024) | ✓ | ✓ | ✗ | ✗ | ✗ | ✗ | ∼405M |
| OMGLLaVA (Zhang et al., 2024) | ✓ | ✓ | ✗ | ✗ | ✗ | ✗ | >220M § |
| GSVA (Xia et al., 2024) | ✓ | ✓ | ✗ | ✗ | ✗ | ✗ | ∼13B |
| ReLA (Liu et al., 2023) | ✓ | ✓ | ✗ | ✗ | ✗ | ✗ | 3B |
| SAM4MLLM (Chen et al., 2024) | ✓ | ✓ | ✗ | ✗ | ✗ | ✗ | 7B |
| ODISE (Xu et al., 2023b) | ✓ | ✓ | ✗ | ✗ | ✗ | ✗ | 28.1M |
| OVDiff (Karazija et al., 2025) | ✓ | ✓ | ✗ | ✗ | ✗ | ✗ | - |
| EmerDiff (Namekata et al., 2024) | ✓ | ✗ | ✗ | ✓ | ✓ | ✓ | - |
| Segment Anyword | ✓ | ✓ | ✓ | ✓ | ✓ | ✓ | **<0.1M**§ |

We report the average trainable parameters, where in normal cases, our Segment Anyword only updates the visual concept corresponded textual embedding parameters.

syntax and dependency structural information.

(4) Extensive experiments on six multi-modal segmentation datasets demonstrate *Segment Anyword establishes a new state-of-the-art performance among training-free methods, even outperforming fully trained or fine-tuned approaches across several metrics*. Furthermore, Segment Anyword generalizes effectively to open-set, reference, and out-of-distribution segmentation, attributed to the novel prompt-learning and alignment design.

## 2. Methods

We start with preliminaries in Section 2.1 and then move to the results of our motivational study in Section 2.2. Section 2.3 describes *Segment Anyword*, a novel visual concept prompt learning framework that leverages a frozen diffu-

sion model for downstream mask prompt adaptation. To improve coherence and completeness of mask prompts, we propose novel regularization techniques that directly integrate linguistic dependency and syntax knowledge for visual prompt binding and clustering (see Section 2.4). Our method demonstrates effortless inverse adaptation to new words and visual concepts with minimal efforts, showcasing its efficiency and adaptability in Figure 2 and Table 1.

### 2.1. Preliminaries

**Text-guided Diffusion Models** are a class of generative models that generate images by progressively denoising random Gaussian noise conditioned on a text prompt. Specifically, we are interested in utilizing a pre-trained denoising network $\epsilon_\theta$. Given an initial random Gaussian noise $\epsilon \sim \mathcal{N}(0, I)$, example image $x$ and text condition $S$, $\epsilon_\theta$ can generate an image $\bar{x} = \epsilon_\theta(\epsilon, V)$ closely resembing $x$. Within $\epsilon_\theta$, the conditioned textual embedding $V = c_\phi(S)$ interacts with the noised image embedding $z_t = \alpha_t z + \sigma_t \epsilon$ via cross-attention layer at time step $t$, where $z = \mathcal{E}(x)$ is encoded from a frozen image encoder $\mathcal{E}$, and $\alpha_t$ and $\sigma_t$ are noise schedulers. The token-wise cross-attention maps, $M = \text{softmax}(QK^\top/\sqrt{d})$, correlate to the similarity between image embedding query $Q = f_Q(z_t)$ and textual embedding key $K = f_K(V)$. In Segment Anyword, we average the cross-attention maps across all time steps to identify high-confidence areas for object localization.

**Inversion** aims to invert a given image $x$ back into the latent space, such that the input image can be faithfully reconstructed from the inverted deterministic latent representation $\boldsymbol{z}^\star = [z^\star_{t=1}, \cdots, z^\star_{t=T}]$. Recent techniques focus on utilizing a textual inversion to regenerate customized images from few concept shots (Gal et al., 2022). We predict $z^\star_t$ based on the presumption that the denoising process can be reversed in the limit of infinite small steps with the noise scheduler $\alpha_t$ (Zhu et al., 2016; Xia et al., 2022):

$$z^\star_t = \frac{\sqrt{\alpha_t}}{\sqrt{\alpha_{t-1}}} z^\star_{t-1} + \sqrt{\alpha_t} \left( \sqrt{\frac{1}{\alpha_t} - 1} - \sqrt{\frac{1}{\alpha_{t-1}} - 1} \right) \epsilon_\theta(z^\star_{t-1}, t-1). \quad (1)$$

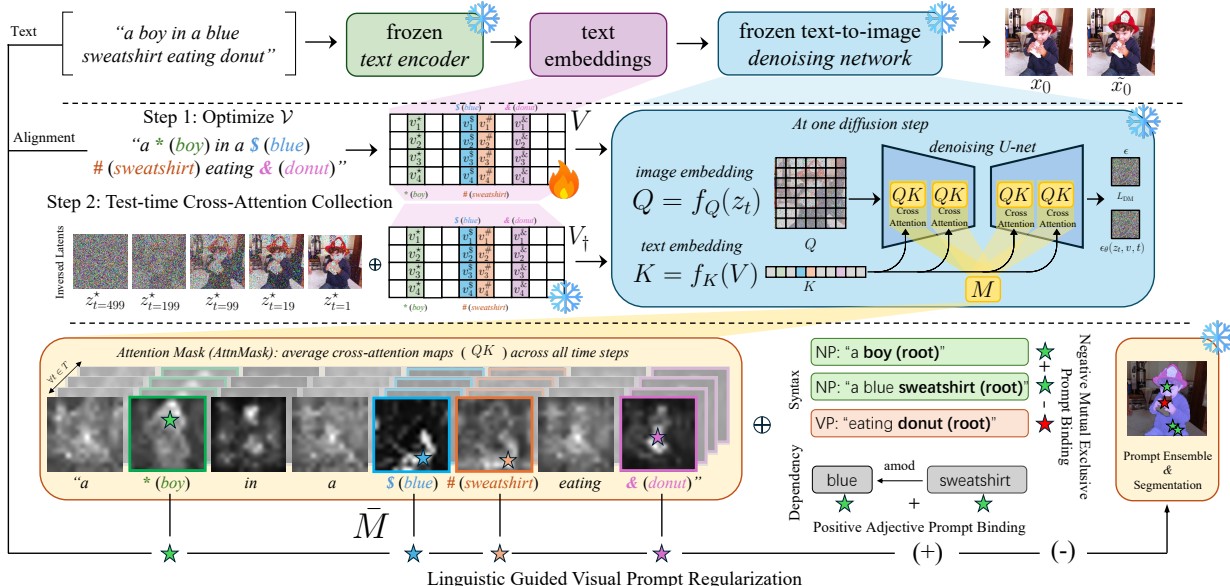

Figure 4. **Pipeline overview.** Segment Anyword leverage the inversed scalability of a frozen text-to-image denosing diffusion model $\epsilon_\theta$ for training-free open-set language grounded segmentation. First, Segment Anyword regards the segmentation reference expression (top-left) as image generation text condition and reconstruct the input image $x_0$. Within image reconstruction process, Segment Anyword only update the textual embedding $V = [v^\star, \ldots, v^{\&}]$ of visual concepts (coloured texts) while the rest of network parameters ($c_\phi, \epsilon_\theta$) remain frozen. At test time, the averaged cross-attention maps are collected through an diffusion process of $Z^\star$ with optimized $V$. We further introduce novel linguistic-guided visual prompt regularization to bind and cluster mask prompts for mining noise-tolerant prompts and improve downstream segmentation accuracy by integrating sentence denpendency and syntax structural information directly.

We utilize direct inversion (Ju et al., 2024) to preserve the original image, where the distance between the inverse and the denoising chains is minimized by the addition operator.

**Segment Anything Model** (SAM) (Kirillov et al., 2023) is a general segmentation model consisting of an image encoder, a prompt encoder, and a mask decoder. Given an image with a set of visual prompts $\boldsymbol{p} = [p_1, \ldots, p_N]$, SAM extracts image features and prompt tokens to interact with mask tokens in a two-way transformer. The final segmentation is generated by multiplying mask tokens with image features. A vanilla SAM cannot automatically control the mask category and granularity for segmented objects, resulting in numerous irrelevant segmentation results. Here we simply apply a frozen SAM as a pure post-processing module by feeding in object location priors as point prompt $\boldsymbol{p} \in \mathbb{R}^{N \times 2}$ from our prompt-learning framework, with automatic label assignment inherited from text expression as a language grounded segmentation. Compared with related works that utilized SAM as mask generator (Chen et al., 2024), our Segment Anyword did not require additional training or fine-tuning, which is computationally lightweight and resource effective.

## 2.2. Motivational Study

To understand the capabilities of existing methods for open-set segmentation, we begin with a large scale motivational

study. We modify the subject noun-phrase in each text reference while preserving the semantic meaning of original sentence to create an open-set learning space. Thus we extend each image associating with additional generated 2-5 mutated expressions (*e.g.* [ apple pieces]→ [ apple pieces, apple slices, cut up apples]). We retain the Intersection-over-Union (IoU) of each image-text pair and calculate the mean (IoU Mean) and Standard Deviation (IoU Std) across all associated expressions. As shown in Figure 3, we observe that: 1) current state-of-the-art VLMs struggle with stable segmentation performance from inaccurate boundaries or mis-localizing the object. 2) such unstable performance arises from a misalignment between visual mask representation and linguistic embedding, leading to confounded segmentation results when text reference expression varies.

## 2.3. Segment Anyword

As illustrated in Figure 4, Segment Anyword is a prompt learning framework, generating localization prior–*Mask Prompts*–for mask segmentation, by utilizing an off-the-shelf diffusion model to retrieve per-token cross-attention maps. Given an input image-text pair, Segment Anyword learns a list of textual embeddings $V = [v^\star, \ldots, v^{\&}]$ initialized from a frozen text encoder $c_\phi$ such as BERT (Devlin, 2018). This optimization is guided by image-level reconstruction from denoising diffusion process, but only make

**Algorithm 1** Segment Anyword (pseudo code)

1: **Input:** image $x$, text expression $s$, pre-trained $\{c_\theta, \epsilon_\theta\}$.
2: **Output:** aligned embeddings $V_\dagger = [v_\dagger^*, \ldots, v_\dagger^\&]$, mask surrogates $\bar{M} = [\bar{m}^*, \ldots, \bar{m}^\&]$.
3: # updating $V_\dagger = [v_\dagger^*, \ldots, v_\dagger^\&]$ with $L_{DM}$
4: **for** $step = 1$ to $S$ **do**
5:     **Encode** $z = \mathcal{E}(x)$
6:     **Compute** $V = [v^*, \ldots, v^\&] = [c_\theta(s^*), \ldots, c_\theta(s^\&)]$
7:     $V_\dagger := \arg\min_V E_{z,V,t}[\|\epsilon - \epsilon_\theta(z_t, V)\|^2]$
8: **end for**
9: # test-time cross-attention collection
10: **Inversion** noisy latents $Z^* = [z_{t=1}^*, \cdots, z_{t=T}^*]$ with Eqn. 1
11: $Z = \text{Denoise}(\epsilon_\theta(Z^*, V_\dagger))$
12: $[m_t^*, \ldots, m_t^\&]_{t=1,\cdots,T} = \text{CrossAttention}(Z, V_\dagger)$
    $\bar{M} := \text{Avg}([m_t^*, \ldots, m_t^\&]_{t=1,\cdots,T})$
13: **Return** $(\bar{M}, V_\dagger)$

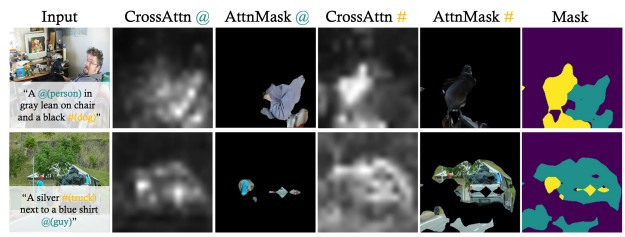

*Figure 5.* Illustration of the averaged cross-attention map and attention mask. By leveraging a frozen denoising text-to-image diffusion network, the word-level cross-attention map offers a location prior. In contrast, the hard mask produces a sub-optimal segmentation result, lacking fine-grained details such as object shape, posture, and boundaries.

$V = [v^\star, \ldots, v^\&]$ as trainable parameter while keeping $c_\phi$ and $\epsilon_\theta$ frozen. This test time embedding updating strategy is a simple yet effective to minimizes the alignment gap between text expression and image. Here, each text expression sentence $s$ can be parsed into a set of noun words of target concepts $\boldsymbol{n} = [n^\star, \ldots, n^\&]$, which are the visual entities to be segmented within the input image. The rest of sentence can be sub-grouped into descriptive words such as adjectives $\boldsymbol{a} = [a^\star, \ldots, a^\&]$ and non-semantic neutral texts $\boldsymbol{t} = [t^\star, \ldots, t^\&]$. We adopt multi-concept textual inversion (Jin et al., 2024) for optimizing $V$ with:

$$L_{DM}(x, \tilde{x}) = E_{z,\epsilon,V,t}[\|\epsilon - \epsilon_\theta(z_t, t, V)\|^2]. \quad (2)$$

Following Equation (1), we invert the input image to generate the deterministic noising latents as test time input. The inverted latents $\boldsymbol{z}^\star$ are sent to a frozen denoising network $\epsilon_\theta$ to reconstruct the original input image. During the denoising process, the updated text embedding $V_\dagger$ interacts with the image features through cross-attention $M$. As illustrated in Figure 4, we regard the high-response attention as the visual concepts activated by the text embedding. We obtain the averaged cross-attention map $\bar{M}$ across all denoising time-steps and sample points from high-response area, as demonstrated in Algorithm 1. The object class label can be retained from the original text expression, where we parse the text expression into Noun-Phrases (NP) and Verb-Phrases (VP) and identify each rooted noun subject (root) within the phrase.

**Limitation of Plain Segment Anyword.** The averaged cross-attention map $\bar{M}$ can be regarded as a pseudo mask, reflecting the token-level correlation between textual prompt and visual object entity. As illustrated in Figure 5, plain Segment Anyword can output $\bar{M}$ as the segmentation mask through a hard threshold. These results indicate that while the cross-attention maps effectively capture the concept-prompt correlation, they lack fine-grained boundary details. This limitation arises from the training objective of diffusion

models, where the network prioritizes image-level reconstruction to achieve a photo-realistic layout of objects, often at the expense of boundary precision as a training shortcut. To address this issue and achieve accurate mask boundaries, we employ a frozen SAM as a post-processing module. Specifically, we get the cross-attention mask from the averaged cross-attention map with a threshold 0.7. A prompt point is then randomly sampled from the largest connected mask region and used as SAM positive mask prompt input.

## 2.4. Linguistics Guided Visual Prompt Regularization

The averaged cross-attention map $\bar{M}$ serves as a localization prior, which is often noisy and coarse. To better mine noise-tolerant visual prompts, we propose two novel prompt regularization strategies that directly guided by sentence syntax and dependency structural information. Some current state-of-the-art methods, such as GLaMM (Rasheed et al., 2024), LISA (Lai et al., 2024), and OMG-LLaVA (Zhang et al., 2024), already incorporate similar linguistic information into their input pipelines such as integrating complete noun phrases during feature fusion or translating a special [**SEG**] token with context into segmentation masks. These models achieve impressive results but require significant training efforts. Other baseline methods focus on test-time optimization, such as CLIPasRNN (Sun et al., 2024), OVD-iff (Karazija et al., 2024), and Peekaboo (Burgert et al., 2022), could theoretically leverage this linguistic information as auxiliary input. However, empirical observations suggest that their performance remains suboptimal under these conditions. In contrast, our proposed method explicitly formalizes linguistic knowledge through prompt regularization. This strategy enables robust mining of noise-tolerant mask prompts, yielding refined and higher-quality segmentation masks, even without extensive training or configuration overhead.

As illustrated in Figure 6, our visual prompt regularization aim to (1) reinforce the correlation between prompts,

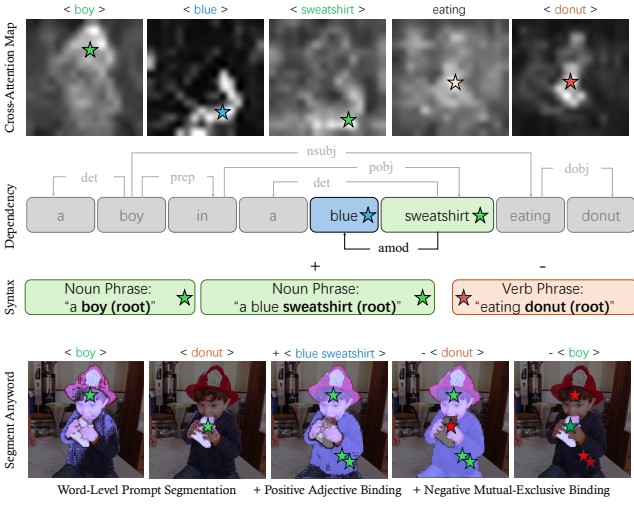

*Figure 6.* **Linguistic guided visual prompt regularization.** Initial visual prompts may lack detailed mask description in both coherence and consistency resulting in suboptimal mask fragments (bottom-left: <boy>, <donut>). Our linguistic guided visual prompt regularization is simple yet effective, leveraging sentence dependency and syntax information. The target object can cluster associated adjective modification word, improving mask completeness (bottom-mid: <sweatshirt> + <blue>). Each entity can be recurrently bind negative prompts from other entities, eliminating false-negative regions for distinct mask boundaries (bottom-right: <boy> + <blue sweatshirt> - <donut>, <donut> - <boy>).

concepts, and masks by clustering positive prompts with descriptive words such as adjectives and (2) eliminate false-negative area by recurrently binding each object with negative prompts from mutual-exclusive entities and their attributes.

**Positive Adjective Prompt Clustering.** Recent research works validate that a pre-trained generative model can also discover descriptive words like adjectives for certain objects and subjects (Rombach et al., 2022; Ruiz et al., 2023; Vinker et al., 2023; Jin et al., 2024). This suggests that leveraging adjective words can reinforce the triplet correlation between the visual concepts, visual prompts, and segmentation masks. For noun words $[n^\star, \ldots, n^\&]$ rooted within parsed noun phrase, the embedding of the corresponding adjective $[a^\star, \ldots, a^\&]$ is also updated during test-time optimization if $[n^\star, \ldots, n^\&]$ is adjective modified by $[a^\star, \ldots, a^\&]$. We retrieve and average the associated cross-attention maps of such adjective across all time steps in denoising process. The adjective cross-attention map follows the same processing pipeline as the noun words to generate a point prompt from its high-activation area. Such prompt from adjective is clustered with the point from noun word as a positive point pair. In cases where the noun word is not accompanied by any adjective words, we sample points twice from the same noun word cross-attention map.

**Negative Mutual Exclusive Prompt Binding.** Positive prompts reinforce the correlation between visual concepts, prompts, and segmentation masks. However, it only operates within the target noun-phrase, which does not inherently promote the mark boundary separation between different noun entities. Leveraging the mutual exclusive relationships between objects, we propose a negative prompt binding strategy guided by sentence syntax structure, where identifiable entities are incompatible with each other at both the linguistic and visual levels. Concretely, for a target noun object, we sample additional points to serve as negative prompts for remaining noun objects. These negative points guide the mask generator by highlighting areas excluded from the current object. In cases where only one object needs segmentation, we randomly sample 1∼3 negative points outside the object mask, which serve as background prompts. By parsing the text expression into noun-phrases, such positive and negative binding strategy can be constructed recurrently, where the clustered positive prompts from mutual exclusive noun-phrase can be grouped as negative prompts for current target as the iteration of noun-phrase progresses.

## 3. Experiments

We perform extensive experiments on six multi-modal image segmentation datasets, including open-set language grounded segmentation dataset GranDf (Rasheed et al., 2024), multi object reference image segmentation dataset gRefCOCO (Liu et al., 2023), single object reference image segmentation dataset RefCOCO, RefCOCO+ and RefCOCOg (Kazemzadeh et al., 2014), open-vocabulary semantic segmentation on Pascal Context (Mottaghi et al., 2014). During evaluation, we follow the official setups for each of the datasets and report: *i)* the AP with a threshold of 50% (AP50), mean IoU (mIoU), and Recall on GranDf, *ii)* the cumulative IoU (cIoU) and generalized IoU (gIoU) on gRefCOCO, and *iii)* the cIoU over all splits on RefCOCO/+/g.

We present implementation details in Appendix C. In addition to accelerate textual embedding update speed, we first use 500 image-text pairs from each dataset training split to LoRA fine-tune BERT text encoder of Segment Anyword with 1100 steps. With LoRA fine-tuned BERT text encoder, Segment can achieve fast textual embedding update within 50 steps during inference on validation and test splits (Segment Anyword$_f$). This step does not require large-scale training or supervision and provides a practical way to efficiently align the text encoder with the target domain, which aligns with our goal of avoiding full training-set traverse and supports fast, sample-efficient test-time optimization.

### 3.1. Main Results

**Open-Set Grounded Segmentation**. GranDf is one of the most challenging datasets in the field of multi-modal

*Table 2.* Open-Set Language Grounded Segmentation Performance on GranDf Dataset validation and test split. "#Images" indicates the number of training images. "-" indicates the result is not reported from original paper.

| Method | #Images | GranDf Val | | | GranDf Test | | |
|---|---|---|---|---|---|---|---|
| | | AP50 | mIoU | Recall | AP50 | mIoU | Recall |
| *Training End-to-End MLLM* | | | | | | | |
| 1. BuboGPT | 130M | 19.1 | 54.0 | 29.4 | 17.3 | 54.1 | 27.0 |
| 2. Kosmos-2 | 9M | 17.1 | 55.6 | 28.3 | 17.2 | 56.8 | 29.0 |
| 3. LISA | 0.12M | 25.2 | 62.0 | 36.3 | 24.8 | 61.7 | 35.5 |
| 4. GLaMM | 11M | 28.9 | 65.8 | 39.6 | 27.2 | 64.6 | 38.0 |
| 5. OMG-LLaVA | 74K | 26.9 | 64.6 | - | 26.1 | 62.8 | - |
| *Fine Tuning MLLM* | | | | | | | |
| 6. GLaMM$_f$ | 200K | 30.8 | 66.3 | 41.8 | 29.2 | **65.6** | **40.8** |
| 7. OMG-LLaVA$_f$ | 200K | 29.9 | 65.5 | - | 28.6 | 64.7 | - |
| *Training-Free Prompt Learning* | | | | | | | |
| 8. Segment Anyword | 0 | **31.3** | **67.4** | 40.7 | 26.6 | 63.4 | 34.7 |
| 9. Segment Anyword$_{f\ w/\ GT\ Text}$ | 500 | 30.2 | 65.9 | **42.4** | **31.1** | 64.1 | 33.9 |

*Table 3.* Multi Object Reference Image Segmentation Performance on gRefCOCO Dataset

| Method | #Image | Val | | TestA | | TestB | |
|---|---|---|---|---|---|---|---|
| | | cIoU | gIoU | cIoU | gIoU | cIoU | gIoU |
| *Traditional methods* | | | | | | | |
| 1. MattNet | 115K | 47.51 | 48.24 | 58.66 | 59.30 | 45.33 | 46.14 |
| 2. LTS | 24K | 52.30 | 52.70 | 61.87 | 62.64 | 49.96 | 50.42 |
| 3. VLT | 24K | 52.51 | 52.00 | 62.19 | 63.20 | 50.52 | 50.88 |
| 4. LAVT | 12K | 57.64 | 58.40 | 65.32 | 65.90 | 55.04 | 55.83 |
| *VLM based methods* | | | | | | | |
| 5. CRIS | 12K | 55.34 | 56.27 | 63.82 | 63.42 | 51.04 | 51.79 |
| 6. ReLA | 12K | 62.42 | 63.60 | 69.26 | 70.03 | 59.88 | 61.02 |
| *MLLM based methods* | | | | | | | |
| 7. LISA | 0.12M | 38.72 | 32.21 | 52.55 | 48.54 | 44.79 | 39.65 |
| 8. GSVA | 0.12M | 61.70 | 63.32 | 69.23 | 70.11 | 60.26 | 61.34 |
| *Fine-tuned methods* | | | | | | | |
| 9. LISA$_f$ | 256 | 61.76 | 61.63 | 68.50 | 66.27 | 60.63 | 58.84 |
| 10. GSVA$_f$ | 16K | 63.29 | 66.47 | 69.93 | 71.08 | 60.47 | 62.23 |
| 11. SAM4MLLM | 110K | 66.33 | **68.96** | 70.13 | 70.54 | 63.21 | 63.98 |
| *Training-Free Methods* | | | | | | | |
| 12. CLIPasRNN | 0 | 16.8 | - | - | - | - | - |
| 13. PSALM (Zero-Shot) | 130M | 42.00 | 43.30 | 52.40 | 54.50 | 50.60 | 52.50 |
| 14. Segment Anyword$_f$ | 500 | **67.73** | 66.08 | **73.57** | **74.63** | **67.56** | **70.90** |

*Table 4.* Single Object Reference Image Segmentation Performance on RefCOCO, RefCOCO+ and RefCOCOg Dataset. The evaluation metric is cIoU.

| Method | #Images | RefCOCO | | | RefCOCO+ | | | RefCOCOg | |
|---|---|---|---|---|---|---|---|---|---|
| | | Val | TestA | TestB | Val | TestA | TestB | Val(U) | Test(U) |
| *Traditional methods* | | | | | | | | | |
| 1. LAVT | 12K | 72.7 | 75.8 | 68.8 | 62.1 | 68.4 | 55.1 | 61.2 | 62.1 |
| *VLM based methods* | | | | | | | | | |
| 2. CRIS | 12K | 70.5 | 73.2 | 66.1 | 65.3 | 68.1 | 53.7 | 59.9 | 60.4 |
| 3. ETRIS | 12K | 71.06 | 74.11 | 66.66 | 62.23 | 68.51 | 52.79 | 60.28 | 60.42 |
| 4. VPD | 12K | 73.25 | - | - | 62.69 | - | - | 61.96 | - |
| *MLLM based methods* | | | | | | | | | |
| 5. LISA | 0.12M | 74.9 | 79.1 | 72.3 | 65.1 | 70.8 | 58.1 | 67.9 | 70.6 |
| 6. GSVA | 0.12M | 77.2 | 78.9 | 73.5 | 65.9 | 69.6 | 59.8 | 72.7 | 73.3 |
| 7. GLaMM | 11M | 79.5 | 83.2 | 76.9 | 72.6 | 78.7 | 64.6 | 74.2 | 74.9 |
| *Training-Free methods* | | | | | | | | | |
| 8. GL-CLIP | 0 | 26.20 | 24.94 | 26.56 | 27.80 | 25.64 | 27.84 | 33.52 | 33.67 |
| 9. CLIPasRNN | 0 | 33.57 | 35.36 | 30.51 | 34.22 | 36.03 | 31.02 | 36.67 | 36.57 |
| 10. Segment Anyword$_f$ | 500 | 55.32 | 47.87 | 66.04 | 55.57 | 47.43 | 67.04 | 58.43 | 60.09 |

*Table 5.* Open-Vocabulary Segmentation Performance PASCAL Context 59 (PC-59) Dataset

| Method | Training Data | PC-59 mIoU |
|---|---|---|
| *Traditional methods* | | |
| 1. OVSegmenter | 4.3M | 20.4 |
| 2. GroupViT | 12M | 23.4 |
| *CLIP based methods* | | |
| 3. SegCLIP | 3M | 24.7 |
| 4. MaskCLIP | 0 | 26.4 |
| *Fine-tuning diffusion based methods* | | |
| 5. ODISE | 83K | 57.3 |
| *Training-free diffusion based methods* | | |
| 6. OVDiff | 0 | 32.9 |
| 7. CaR | 0 | 39.5 |
| 8. EmerDiff | 0 | 45.7 |
| 9. Segment Anyword | 0 | 52.5 |

image segmentation, as it includes multi-domain images from MSCOCO (Lin et al., 2014), PSG (Yang et al., 2022a), and SA-1B (Kirillov et al., 2023), along with detailed manual text descriptions. This diversity presents significant challenges in understanding and aligning multi-modal information, matching object segmentation with associated noun-phrase. Our proposed Segment Anyword demonstrates strong perception capabilities compared to comprehensive baseline methods. As shown in Table 2, our approach outperforms most recent MLLM methods and achieves results comparable to state-of-the-art models such as GLaMM (Rasheed et al., 2024) and OMG-LLaVA (Zhang et al., 2024). Notably, we establish new state-of-the-art results in AP50 and mIoU on the GranDf validation set. Compared to GLaMM, which also employs SAM as a post-processing module, our method benefits from grounded visual concept prompt learning, effectively preserving vision-language inference capabilities.

**Reference Image Segmentation**. We also evaluate Seg-

ment Anyword on reference image segmentation, which aim to segment binary mask of text referred objects in image without requiring grounded mask-phrase mapping. We report multi-object reference image segmentation on gRef-COCO dataset in Table 3 and single-object reference image segmentation on RefCOCO, RefCOCO+ and RefCOCOg in Table 4. Following the original setup of GLaMM, we perform cross-matching between predictions and ground truths based on IoU. Compared with zero-shot methods including CaR (Sun et al., 2024) and PSALM (Zhang et al., 2025), we achieve new state-of-the-arts on gRefCOCO across all splits, ranging from 18.40 gIoU to 25.73 cIoU. Despite fine-tuned methods such as SAM4MLLM (Chen et al., 2024) utilizing SAM to extract object localization information, Segment Anyword demonstrates superior performance with a notable margin of 1.40 cIoU to 6.92 gIoU. These results highlight that by simply leveraging visual concept prompts from cross-attention maps generated by a frozen diffusion model, our method effectively bridges discriminative segmentation tasks with generative models, offering significant advantages in designing perceptual systems capable of handling complex task such as simultaneously referring and segmenting multiple objects.

*Table 6.* Ablation Study of core components of Segment Anyword on GranDf dataset Validation Split. **PL:** test-time prompt learning. **R1:** positive adjective prompt clustering from dependency regularization. **R2:** negative mutual-exclusive prompt binding from syntax regularization.

| Components | | | | GranD Validation | |
|---|---|---|---|---|---|
| w/SAM | PL | R1 | R2 | mAP | mIoU |
| ✗ | ✗ | ✗ | ✗ | 8.2 | 13.7 |
| ✓ | ✗ | ✗ | ✗ | 16.9 | 32.4 |
| ✓ | ✓(550 steps) | ✗ | ✗ | 19.1 | 38.8 |
| ✓ | ✓(1100 steps) | ✗ | ✗ | 22.1 | 42.6 |
| ✓ | ✓(1100 steps) | ✓ | ✗ | 24.9 | 62.2 |
| ✓ | ✓(1100 steps) | ✗ | ✓ | 28.5 | 63.1 |
| ✓ | ✓(1100 steps) | ✓ | ✓ | 31.3 | 67.4 |
| ✓ | ✓(LoRA+50 steps) | ✓ | ✓ | 30.2 | 65.9 |

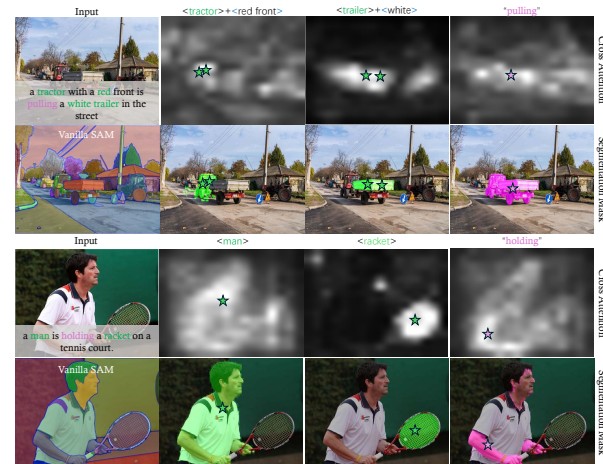

*Figure 7.* Predicate Mask Visualization. For predicate words without explicit semantic meaning, Segment Anyword can be prompted to identify the predicate masks, such as (a) a relation between subject and object entities and (b) human-object interaction.

**Open-Vocabulary Semantic Segmentation**. We also compare Segment Anyword with previous open-vocabulary segmentation methods using Pascal Context 59 (PC59), which contains 59 object categories (Table 5). Segment Anyword reports a new state-of-the-art result, surpassing other diffusion based training free methods such as OVDiff (Karazija et al., 2024) and EmerDiff (Namekata et al., 2024) by a largin margin (19.6 and 6.8 mIoU). This result validates that without generating synthetic support image sets or storing intermediate features, retrieve visual prompts from cross-attention map from a frozen diffusion model can still achieve superior segmentation result. Despite ODISE (Xu et al., 2023b) trained additional mask encoder and decoder with large number of samples, Segment Anyword shows comparable semantic segmentation capability. which provides a feasible solution for deploying training free open-vocabulary semantic segmentation methods in low resource conditions.

### 3.2. Ablation Study

We perform ablation studies to fully understand each factor contribution using GranDf Validation split. We focus on the best performing variant, Segment Anyword with textual embedding update (PL), positive adjective prompt clustering from dependency regularization (R1), negative mutual-exclusive prompt binding from syntax regularization (R2), promptable segmentor post-processing (w/SAM) and text encoder LoRA fine-tune for fast textual embedding updating within 50 steps.

**SAM Post-processing**. We first validate the boosted performance from SAM as a post-processing module. We examine the proposed method, Segment Anyword, which uses a thresholded cross-attention mask as the segmentation output. We show a boosted segmentation performance improved from 13.7 to 32.4 mIoU. This verifies our observation of plain Segment Anyword, where the cross-attention map can capture the concept-prompt correlation but lacks fine-grained details at object mask boundaries, leading to

a suboptimal performance for segmentation task, which requires pixel-level mask delination.

**Textual Embedding Update**. We validate the contribution of test-time textual embedding updating, which only updates a very small number of parameters for aligning token embedding with image features. As reported in Table 6, we obtain 6.4 mIoU increase with 550 optimizing steps and further 10.2 mIoU increase with 1100 optimizing steps. This result validates the findings from our motivational study that without test-time textual embedding optimization, vanilla Segment Anyword suffers from terminology variance. We also validate that using LoRA to fine-tune text-encoder can achieve fast text domain adaptation, decreasing the updating steps of textual embedding updating from 1100 to 50 while preserving mask surrogates effectiveness for downstream promptable segmentation, providing a reasonable trade-off between inference speed and accuracy.

**Linguistic Guided Visual Prompt Regularization**. We explored the influence of linguistic guided visual prompt regularization. As reported in Table 6, Segment Anyword$_{PL-R1-R2}$ achieves best performance with final 31.3 AP50 and 67.4 mIoU on GranDf validation split. The results suggest that injecting linguistic information, both dependency structure and syntax structure, can generate a robust visual prompt combination with best capability for noisy tolerance, which helps promptable segmentation generator such as SAM eliminate false-positive areas, bind inclusive object attribute fragments and generate boundary distinct segmentation masks.

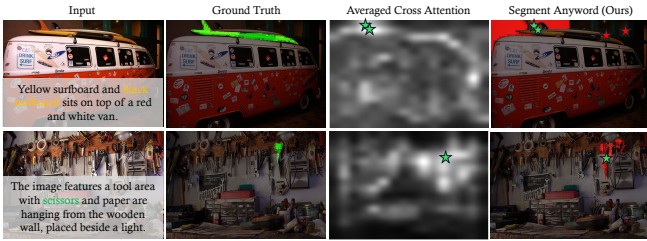

**Figure 8.** Failure Case Visualization. Our method struggles with very tiny objects composed with thin structures, due to restricted cross-attention resolution.

### 3.3. Predicate Segmentation

Language-driven visual concept discovery is a human-computer interaction where a user describe an image using multiple unfamiliar concepts. The entities of interest are often concrete linguistic subjects and objects, with explicit visual geometric and semantic characteristics. In contrast, predicate words such as verbs or gerunds with abstract meanings, have been less explored in previous multi-modal segmentors. However, predicate words are essential for linking, reasoning, and understanding multi-modal information. Without training or tuning additional modules, we demonstrate that *Segment Anyword can also effectively learn a transferable visual prompts for abstract predicate words.* As shown in Figure 7, Segment Anyword can not only prompt concrete object masks but also reveal the correlation between object-object (**'pulling'** a trailer) and human-object (**'holding'** a racket). To the best of our knowledge, *Segment Anyword is the first approach capable of handling both concrete and abstract visual concepts in open-set segmentation.* By learning predicate correlations representations of visual entities, it can reduce hallucinations in image generation and editing. Furthermore, it holds great potential to accelerate scientific knowledge discovery by learning entity relationship from experimental observations or textbooks.

### 3.4. Failure Case Analysis

Although Segment Anyword achieves promising result, we acknowledge that it can still encounter failure segmentation. Similar to other image segmentation methods, Segment Anyword struggles to segment tiny objects with tiny structures (Figure 8), as the averaged cross-attention offers limited detail for precise shape contours due to its restricted resolution size (16×16). Analyzing Segment Anyword with a high-resolution backbone is a promising direction in future.

### 4. Conclusion

We introduce Segment Anyword for open-set language grounded segmentation. Our motivational study reveals that current VLMs or MLLMs struggle to achieve stable perfor-

mance as the text reference varies. In contrast, Segment Anyword is a novel visual concept prompt learning framework that leverages a frozen diffusion model leveraging intermediate cross-attention maps with optimized texutal embedding to localize token-level visual concepts for promptable segmentation. We propose novel prompt binding and clustering regularization that enable the direct transfer of linguistic structure knowledge into visual prompts, which enhancing the effectiveness of the mask prompts with dependency and syntax information. Extensive experiments across different tasks and datasets demonstrate our approach's superior performance, confirming its robustness and adaptability.

### Acknowledgment

We thank the area chair and reviewers for their constructive feedbacks. Most experiments were performed using the Sulis Tier 2 HPC platform hosted by the Scientific Computing Research Technology Platform at the University of Warwick. Sulis is funded by EPSRC Grant EP/T022108/1 and the HPC Midlands+ consortium.

### Impact Statement

This paper presents work whose goal is to advance the field of Machine Learning. There are many potential societal consequences of our work, none which we feel must be specifically highlighted here.

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

# A. Additional Motivational Study Results

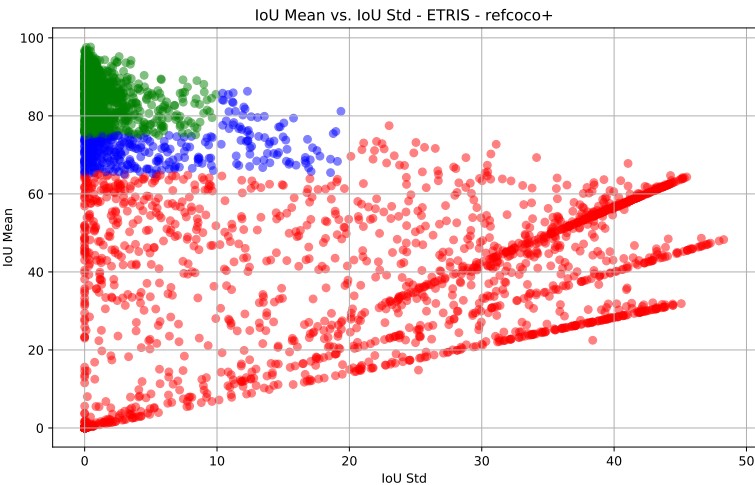

*Figure 9.* **Motivational Study** on the state-of-the-art parameter efficient fine-tuning model ETRIS (Xu et al., 2023c) for reference segmentation.

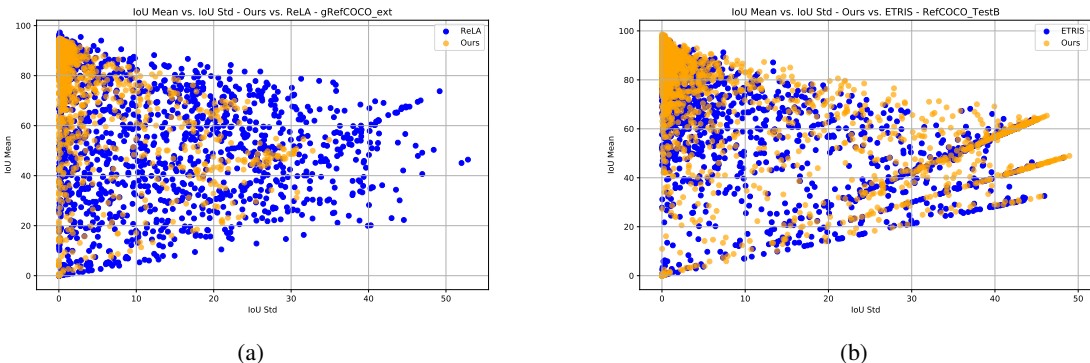

(a)                                                                                        (b)

*Figure 10.* (a) Scatter plot of IoU mean versus IoU standard deviation per image sample on the gRefCOCO dataset against ReLA. (b) Scatter plot of IoU mean versus IoU standard deviation per image sample on the RefCOCO+ dataset against ETRIS. Yellow dots represent results from our method, while blue dots represent results from baseline methods. The plots show that by focusing on test-time optimization, our method effectively improves segmentation accuracy and maintains stability under diverse textual expressions, pushing the results toward the top-left corner.

We present an additional motivational study using another state-of-the-art model, ETRIS (Xu et al., 2023c), in Fig. 9. ETRIS focuses on aligning representations from pre-trained visual and language encoders by introducing intermediate fine-tuned adapters. The results further validate our previous findings: without test-time alignment, current open-set segmentation models are vulnerable to input variance, leading to unstable segmentation results.

As demonstrated through comprehensive experiments and additional results in Fig. 10, our method effectively improves both accuracy and stability by leveraging an off-the-shelf diffusion model to extract mask prompts via an inversion process. This approach is modular and pluggable, offering several advantages including the ability to handle novel visual and linguistic concepts without requiring additional supervision or complex training procedures.

# B. Related Works

**Vision Language Representation Learning** Learning a generalized and transferable vision-language representation has been extensively explored, driven by particular tasks such as visual question answering (Antol et al., 2015) and recently rocket-rising Artificial Intelligence Generated Content (AIGC) (Ramesh et al., 2021; Rombach et al., 2022). Building on the advancements of pre-trained language models (Kenton & Toutanova, 2019; Radford, 2018), large-scale VLMs were proposed for training with image-text pair contrastive learning (Radford et al., 2021) over large-scale multi-modal datasets (Sun et al., 2017; Thomee et al., 2016). Such strong VLMs transformed the paradigm of visual task setting, with task complexity increasing from pre-defined close-set categorical labels to free-form open-set natural language references (Zhou et al., 2024). Recent works such as CLIPpy (Ranasinghe et al., 2023) and PACL (Mukhoti et al., 2023) assign labels by contrasting image patch embeddings with object text label embeddings. However, these methods still require specific training configurations. Peekaboo (Burgert et al., 2022) is closely related to our approach, as both aim to learn visual concepts using off-the-shelf diffusion models. However, Peekaboo heavily relies on alpha map initialization and is sensitive to its quality. Other seminal works in prompt learning, such as CoOp (Zhou et al., 2022d), introduced prompt tuning to adapt vision-language models like CLIP for few-shot tasks. However, CoOp is limited when handling unseen classes. Its successor, CoCoOp (Zhou et al., 2022c), extended this approach by providing input-conditioned adaptability, making it more suitable for open-world segmentation tasks involving diverse textual expressions. While CoCoOp demonstrates robustness under rigorous testing, it introduces additional computational complexity.

While the results are promising, our motivational study described in Section 2.2 highlights that discovering and aligning visual concepts between dense pixel-level mask representations and word-level text embeddings remains challenging, leading to unstable and confounded segmentation results.

**Open-Set Image Segmentation.** Unlike traditional image segmentation which operates within a close-set pre-defined labels, open-set image segmentation requires recognizing previously unseen objects and categories described using novel words or sentences (Zhu & Chen, 2024; Wu et al., 2024), which is essential for real-world applications where adaptability to diverse terminology is paramount. Early efforts leveraged powerful pre-trained VLMs such as CLIP (Radford, 2018) to bridge the gap between visual and textual modalities by contrastive learning, prototype matching, and similarity clustering (Zhou et al., 2022b; Wang et al., 2022; Xu et al., 2023b;c). Although effective to some extent, they often fall short in handling complex and descriptive sentences encountered. Recent advancements introduced multi-modal large language models to improve the understanding of natural language descriptions and decoding final mask tokens (Lai et al., 2024; Rasheed et al., 2024; Xia et al., 2024; Zhang et al., 2024). However, these approaches require substantial computational costs for training or fine-tuning, where performance deteriorates when faced with out-of-vocabulary words not covered by pre-trained models. Methods including (Xia et al., 2024; Chen et al., 2024) can generate multiple object masks within a single binary segmentation map. However, these masks are not explicitly associated with specific word indices. Moreover, neither model is capable of handling novel concepts, as both require training or fine-tuning on the target dataset. This limits their ability to generalize to unseen concepts during test time. In practical applications, text references tend to be highly complex and detailed as users provide intricate descriptions to guide segmentation tasks (Eger et al., 2019; Le et al., 2023). To overcome these limitations, we introduce Segment Anyword, a novel framework that adaptively aligns to diverse terminologies to describe unified visual concepts or objects. By leveraging the inverse scalability of an off-the-shelf pre-trained diffusion model, Segment Anyword achieves superior adaptability with minimal visual prompt optimization effort, offering a practical and efficient solution for open-set segmentation.

**Diffusion Models for Image Segmentation.** Diffusion models have significantly advanced image generation, recognized for their proficiency in modeling data distributions through forward noising and backward denoising processes (Ho et al., 2020). Beyond powerful generative capabilities, diffusion models offer valuable insights into discriminative tasks such as image classification and segmentation (Luo et al., 2024; Meng et al., 2024). Previous works, such as EmerDiff (Namekata et al., 2024), DAAM (Tang et al., 2023) and OVAM (Marcos-Manchón et al., 2024), have successfully detected objects by identifying salient perturbations in attention maps. However, they often struggle with robust language grounded object instantiation. Other approaches, such as VPD (Zhao et al., 2023a), have sought to enhance segmentation by fine-tuning a mask decoder on a pre-trained diffusion network, which requires resource-intensive pixel-wise annotations. Zero-shot methods like OVDiff (Karazija et al., 2024) and DiffusionSeg (Tian et al., 2023) typically require intricate post-processing steps, including prototype matching and iterative refining. In contrast, Segment Anyword employs a frozen text-to-image diffusion model that operates without additional training or fine-tuning, which is not only more computationally efficient, simpler, and more straightforward, but also remains highly effective.

# C. Dataset and Implementation

## C.1. Implementation Details

In our motivational study, we generate the mutated expressions by querying ChatGPT4o, with a template prompt as follows:

> Prompt:
>
> *As a NLP expert, please genrate a list of n synonyms of the noun phrases in the following [sentence] and output the list separated by '&'*

where *n* is in **randint**(2, 5) and [*sentence*] is set to original labeled text referring expression.

Unless specified, we retain the original hyper-parameter of latent diffusion model (LDM) (Patashnik et al., 2023) as our diffusion backbone. We follow the visual concept discovery process specified in multi-concept prompt learning (MCPL) (Jin et al., 2024). Differently from MCPL that uses pseudo placeholder for concept learning, we use the pre-defined text expression from the data as a condition prompt where all nouns, adjectives and prepositions are explicitly described by human annotator. Our experiments were executed on a single 40G A100 GPU with a batch size of 8. The base learning rate for textual embedding was set to $0.005$. The hyper-parameters of textual embedding updating remains the same in LDM and MCPL, with the temperature and scaling term $(\tau, \gamma)$ of $(0.3, 0.00075)$. We use BERT (Devlin, 2018) to generate token embeddings. For words included in BERT's pre-trained vocabulary, we directly use their pre-trained embeddings. For out-of-vocabulary words, token embeddings are randomly initialized. By leveraging the pre-trained token embedding initialization, we reduce the optimization steps to 1,100, achieving a speedup of $6\times$ compared to LDM and MCPL. To accelerate inference time textual embedding update speed, we further introduce LoRA fine-tuned text encoder for fast inference time textual embedding update (Segment Anyword$_f$). For each evaluation dataset, we randomly sample 500 image-text pairs from training set to perform a LoRA fine-tune on BERT encoder only, with LoRA hyper-parameter $r = 16$. With LoRA fine-tuned BERT text encoder, Segment Anyword$_f$ achieve a fast inference time text domain adaptation, decreasing textual embedding update steps from 1100 to 50 by sacrificing a relative small accuracy.

We choose the fine-tuned version of Vicuna-7B-v1.5 (Zheng et al., 2023) as our large language model (LLM) to parse the text prompt and generate the noun phrases, which is adopted by a previous state-of-the-art grounding segmentation method (Rasheed et al., 2024) with a low-rank adaptation scale $\alpha = 8$. We keep LLM parameters frozen. For noun phrases parsed from the original text prompt, we use part-of-speech tagging to identify the root noun as the object to be segmented and the adjective modification (amod) as the positive prompt binding.

We use cross-attention maps at the 16×16 resolution, averaged across all denoising time steps, to obtain the final cross-attention. This setup follows prior works such as MCPL (Jin et al., 2024) and Prompt2Prompt (Hertz et al., 2023), ensuring a fair and consistent implementation for updating textual embeddings. For the post-processing module, we utilize a frozen SAM with ViT-H as the promptable mask generator.

## C.2. Dataset Details

**GranDf** (Rasheed et al., 2024) is motivated by the need for higher-quality data during the fine-tuning stage. It comprises 214K image-grounded text pairs, along with 2.5K validation samples and 5K test samples, sourced from an extension of open-source datasets, including Flickr-39K (Plummer et al., 2015), RefCOCOg (Kazemzadeh et al., 2014), and PSG (Yang et al., 2022a). The original aim of GranDf is for Grounded Conversation Generation (GCG), involving both reference generation and grounded image segmentation. As a segmentation model, we only focus on segmentation capability evaluation by using ground truth text expression as segmentation reference.

**RefCOCO** (Kazemzadeh et al., 2014) dataset contains 142,210 referring expressions for 50,000 objects across 19,994 images. Collected from MSCOCO (Lin et al., 2014) through a two-player game, it is divided into 120,624 training, 10,834 validation, 5,657 testA, and 5,095 testB samples. On average, each referring expression has a mean length of 3.6 words.

**RefCOCO+** (Kazemzadeh et al., 2014) dataset consists of 141,564 referring expressions linked to 49,856 objects in 19,992 images. It follows the same train-validation-test split as RefCOCO, with 120,624 training, 10,758 validation, 5,726 testA, and 4,889 testB samples. Unlike RefCOCO, RefCOCO+ excludes absolute-location words, making it a more challenging benchmark for referring image segmentation with averaged sentence length of 3.53 words.

**RefCOCOg** (Mao et al., 2016) comprises 104,560 referring expressions for 54,822 objects across 26,711 images. Unlike RefCOCO and RefCOCO+, its natural expressions were collected from Amazon Mechanical Turk, resulting in longer and more descriptive phrases, averaging 8.4 words per expression. Additionally, RefCOCOg contains more detailed location and appearance-based descriptions, and in this work, we adopt the UNC validation and test partition for evaluation.

**gRefCOCO** (Liu et al., 2023) dataset comprises 278,232 referring expressions associated with 19,994 images, including 80,022 multi-target and 32,202 empty-target expressions. Following the UNC partition of RefCOCO, the dataset is divided into training, validation, testA, and testB subsets. The validation set contains 1,485 images with 5,324 sentences, while testA includes 750 images with 8,825 sentences, and testB consists of 749 images with 5,744 sentences.

**PASCAL Context** (Mottaghi et al., 2014) dataset is an extension of the PASCAL VOC 2010 detection challenge, with pixel-level segmentation mask annotation. We used a subset, which contains 59 frequent classes (PC59) for evaluation, with 5,100 images in validation set.

# D. Competing Baselines

Below we present related works we selected as comparison baselines, where we focusing on comparison with previous state-of-the-art methods, also fair comparison based on same training-free pipeline, SAM as mask refine module and frozen text-to-image diffusion model as backbone. For task definition, please refer Figure 11.

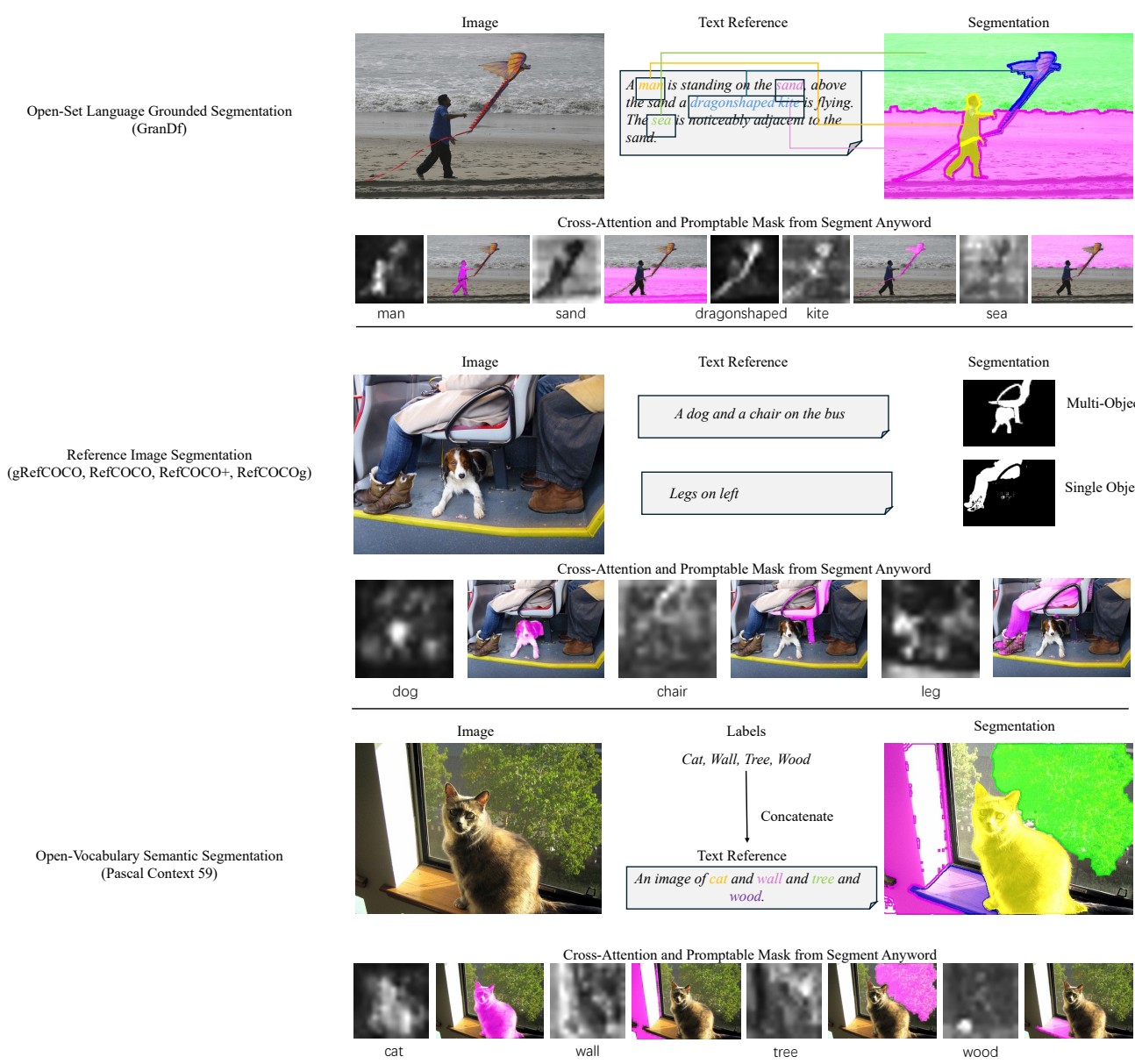

*Figure 11.* **Open-set Image Segmentation Comparison.** From top to bottom, the sentence complexity decrease from free-form descriptive expression to key-word concatenation. Note that some sentence description in RefCOCO could be ambiguity, as the dataset designs for implicit reasoning and referring.

## D.1. Open-Set Image Segmentation

- BuboGPT (Zhao et al., 2023b) is one of the earliest attempts working on multi-model open-set grounded image segmentation. By fine-tuning on a novel instruction dataset, BuboGPT achieves fine-grained object-level understanding in both conversation generation and grounded recognition using an entity integrated text prompt template '⟨ List of Entities ⟩, ⟨ List of Text ⟩'.

- Kosmos-2 (Peng et al., 2023) enables the perception of object annotation text into visual understanding, with a format string instruction template 'it ⟨ List of annotations ⟩, ⟨ List of Text ⟩'. Kosmos-2 advances the multi-modal understanding capability with novel object location tokens for various downstream tasks including grounded segmentation.

- LISA (Lai et al., 2024) further improve MLLM learning and reasoning with novel question-answering instruction template, predicting the final object segmentation mask token as an answer token 'Sure, it is ⟨ SEG ⟩'. Trained on datasets from both semantic segmentation, reference image segmentation and visual question answering, LISA showsing promising results on domain generalization and reasoning segmentation with world knowledge.

- GLaMM (Rasheed et al., 2024) further improves MLLM understanding by accommodating both text and visual prompts. GLaMM added region level encoders and decoders trained on GranD dataset to extract mask information during MLLM training and fine-tuning.

- OMG-LLaVA (Zhang et al., 2024) extends MLLM reasoning capability at pixel-level by integrating image information, perception priors, and visual prompts into visual tokens. OMG-LLaVA achieves universal segmentation with a fine-tuned multi-modal projector and visual decoder.

## D.2. Reference Image Segmentation

- MattNet (Yu et al., 2018) decomposes the reference expression into three submodules relates to object appearance, location and relationship using a Bi-LSTM, improving localization and segmentation performance.

- LTS (Jing et al., 2021) extract the textual embedding through a GRU and multiply the bottleneck layer image feature within a ConvNet, mimic the perceptual process of 'Locate the Segment'.

- VLT (Ding et al., 2021) trained a transformer based vision-language matching network to query reference expression with image features.

- LAVT (Yang et al., 2022b) replacing the backbone with a vision transformer and inject textual embedding from BERT with attention module.

- CRIS (Wang et al., 2022) utilize CLIP model to contrast reference expression and semantic information by training a multi-modal projector with text-to-pixel constrastive learning.

- ETRIS (Xu et al., 2023c) utilize CLIP model as backbone. In addition, it trained an additional bridging module between vision encoder and text encoder to minimize the alignment gap between multi-modal representations.

- ReLA (Liu et al., 2023) is the first model for multi object reference segmentation framework featureing region-to-text cross-attention.

- GSVA (Xia et al., 2024) utilized a fine-tuned MLLM for reference segmentation. A segmentation mask decoder generates the segmentation mask with joint input from mask token and image features.

- SAM4MLLM (Chen et al., 2024) provide a detailed instruction template for MLLM using SAM to filter object-of-interests, integrating location information into textual template '⟨ List of BBox ⟩'.

- GL-CLIP (Yu et al., 2023) injecting the global and local features using mask proposals for both image and textual embedding, where the final mask the filtered by the cosine-similarity based feature matching.

- CLIPasRNN (Sun et al., 2024) recurrently filter out in irrelevant visual concepts without any training or fine-tuning which preserves the original vocabulary space of off-the-shelf model.

- PSALM (Zhang et al., 2025) extends the MLLM with expert crafted input with images, task instructions, conditional prompts, and mask tokens, which shows promise zero-shot generalization on multi-object reference segmentation.

- VPD (Zhao et al., 2023a) adopts diffusion model as backbone and trained an additional task-specific decoder to generate output based on intermediate features from diffusion model as input.

### D.3. Open-Vocabulary Semantic Segmentation

- OVSegmentor (Xu et al., 2023a) trains a multi-modal transformer to jointly learning image features with raw caption, masked caption and prompted entity for open-vocabulary semantic segmentation.

- GroupViT (Xu et al., 2022) clusters vision transformer intermediate features into superpixel representations, which is further contrastive learned with textual embedding for open-vocabulary segmentation.

- ODISE (Xu et al., 2023b) is one of the earliest attempts utilizing a frozen diffusion model for open-vocabulary segmentation, where the frozen diffusion model generates mask proposals, which is optimized by matching masked-out object features with textual embeddings of object labels.

- MaskCLIP (Zhou et al., 2022b) is one of the earlies attempts proposing train-free and tuning-free methods for open-vocabulary semantic segmentation. MaskCLIP reorganizing CLIP text encoder as a classifier to filter image embedding for a specific object label.

- SegCLIP (Luo et al., 2023) utilize pre-trained CLIP model to learn and contrastive aggregate image patches into superpixels for image reconstruction, which facilities CLIP model with pixel-level understanding.

- OVDiff (Karazija et al., 2024) take advantage of promising generation capability of diffusion model to generate synthetic support set of images for a target object to be segmented, where the object feature are filtered by support set image feature and textual embedding.

- EmerDiff (Namekata et al., 2024) leveraging the intermediate features of a frozen diffusion model, where the segmentation mask are first constructed by k-means clustering feature maps and further refined by evaluating with semantic correspondence through a feature-level perturbation operation.

# E. Additional visualization on Suboptimal Mask from Plain Segment Anyword

Below we present additional visualization on suboptimal segmentation mask from plain Segment Anyword. Through text-to-image reconstruction, we can observe that token-level cross-attention map reflects the correlation between visual concepts and textual representation. We can obtain a naive cross-attention mask through a hard threshold (here is 0.5) as initial segmentation surrogates. Such mask is often coarse and noisy, lacking fine-grained object shape and boundary refinement. Thus we regard such cross-attention as a well candidate for downstream promptable segmentation, where we can randomly sample points as object location prompts, further mined by our novel linguistic guided dependency and syntax structural information regularization.

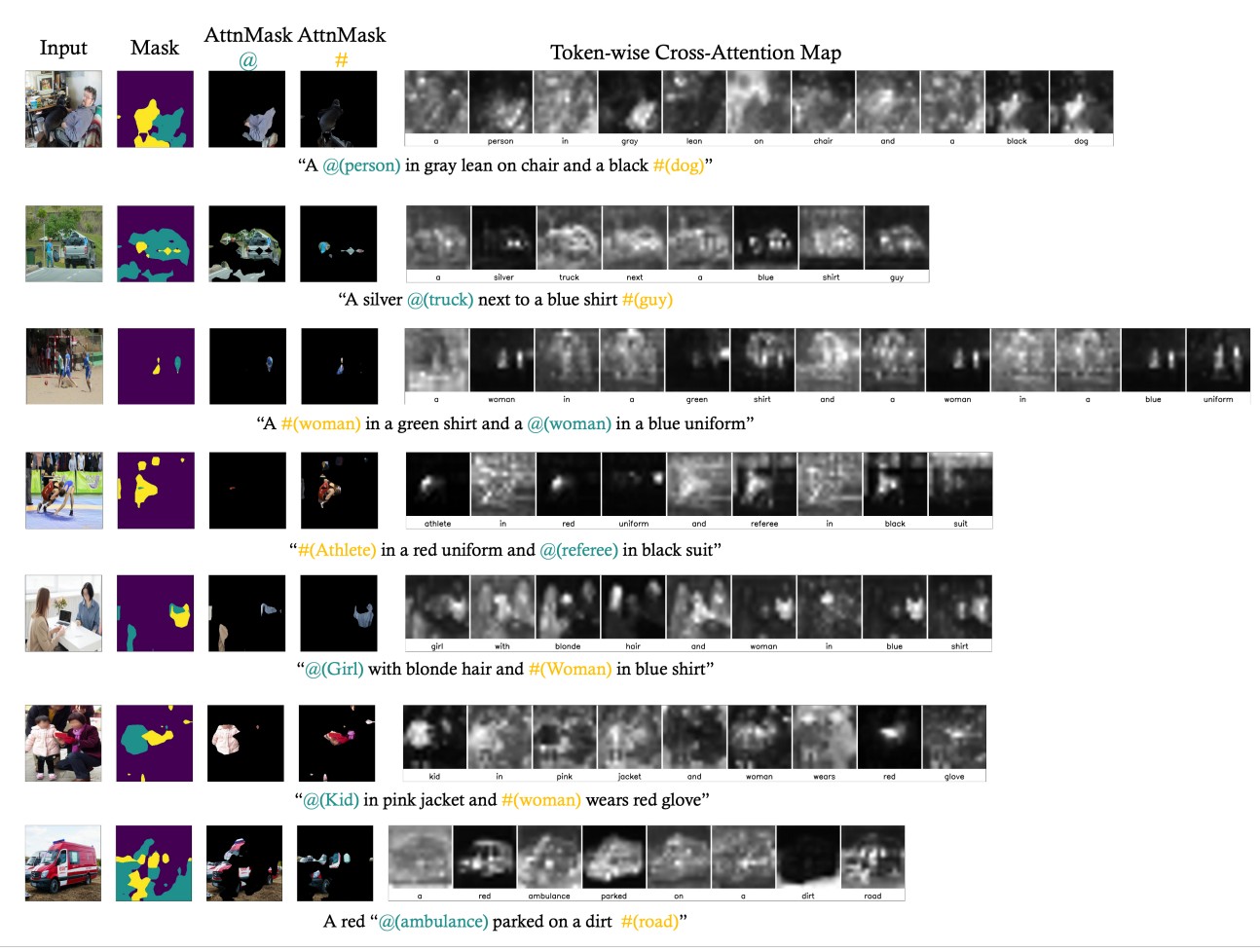

Figure 12. **Additional Suboptimal segmentation masks from plain Segment Anyword.** We show that by reconstructing the input image, Segment Anyword can leverage the cross-attention map from frozen diffusion model as a localization prior to generate mask surrogates. However, a hard threshold cross-attention mask suffer from inaccurate shape recognition and boundary delination.

# F. Additional Ablation Study Results

## F.1. Ablation study on pos-tagging methods

The primary focus of the proposed Segment Anyword is to improve the quality of automatic mask prompts without relying on complex training configurations. The external language model is merely used to parse and index object-related words for cross-attention map retrieval. Importantly, the use of a fine-tuned LLM (e.g., Vicuna (Chiang et al., 2023)) is not required—our method is compatible with state-of-the-art language models such as GPT-4o, which can provide strong reasoning capabilities out of the box. Additionally, standard NLP libraries such as NLTK and SpaCy can be used for text pre-processing as well. These tools are widely adopted in prior work, known to be fast and reliable.

The primary focus of the proposed Segment Anyword is to enhance the quality of automatic mask prompts without relying on complex training configurations. The external language model is used solely to parse and index object-related words for cross-attention map retrieval. Notably, a fine-tuned LLM (e.g., Vicuna (Chiang et al., 2023)) is not required—our method is compatible with state-of-the-art language models such as GPT-4o, which offer strong reasoning capabilities out of the box. Additionally, standard NLP libraries such as NLTK and SpaCy can be used for text pre-processing. These tools, widely adopted in prior work, are known for their speed and reliability.

We conducted the study using 100 randomly selected image-text pairs from RefCOCO (Kazemzadeh et al., 2014). For the SpaCy[1], we used the en_core_web_trf pipeline based on RoBERTa (Liu et al., 2019). We filtered tokens with pos-tag **NOUN** and **ADJ**, and indexed the token with the **nsubj** dependency label as the referred object. For GPT-4o and Vicuna-7B, we used the following prompt:

> Prompt:
>
> *As a NLP expert, you will be provided a caption describing an image. Please do pos tag the caption and identify the only one referred subject object and all adjective attributes. Your response should be in the format of "[(attribute1, attribute2, attribute3, ...), object1]"*
>
> Conditions:
>
> (1) *If the attribute is long, short it by picking one original word.*
>
> (2) *Please include one original word possessive source into the attributes for the subject.*

| Parsing methods | mIoU |
|---|---|
| GPT4o | 68.2 |
| SpaCy(RoBERTa) | 46.9 |
| Vicuna-7B | 59.7 |

*Table 7.* Segmentation results of our Segment Anyword accompanied with different pos-tagging tools.

We present the final segmentation results in Table 7. While SpaCy offers fast, offline parsing, it often misses key adjectives such as color terms like "white." In contrast, GPT-4o provides more accurate parsing, reliably capturing fine-grained attributes. Based on our empirical observations, we offer the following recommendation: for large-scale processing where speed is critical, static NLP libraries such as SpaCy are more suitable due to their efficiency. However, for detailed interactions involving concept learning and prompt refinement, advanced language models like GPT-4o are preferred for their superior reasoning and parsing capabilities.

---

[1]https://github.com/explosion/spaCy

## F.2. Ablation study using llm generated text description

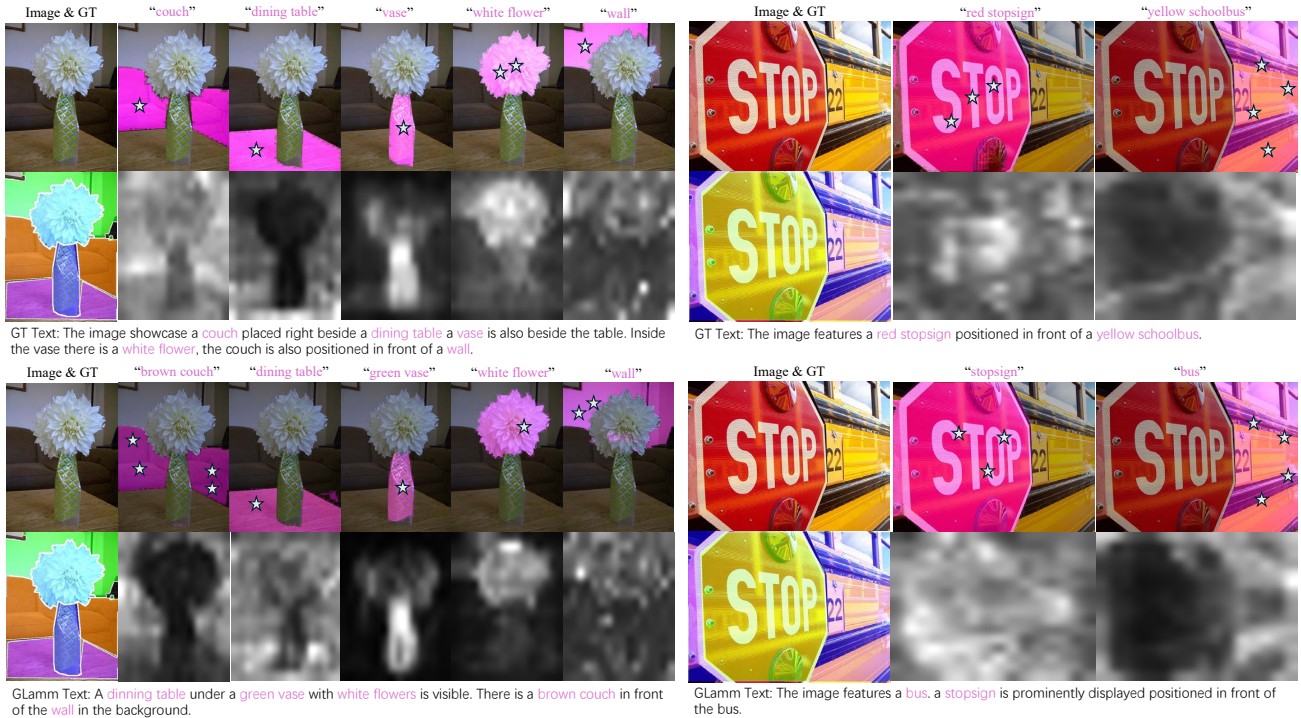

*Figure 13.* Qualitative result comparison between Segment Anyword using ground truth text description and GLaMM generated text description.

| Segment Anyword $_f$ | GranDf Val | | |
|---|---|---|---|
| | AP50 | mIoU | Recall |
| w/ GT Text | 30.2 | 65.9 | 42.4 |
| w/ GLamm Text | 27.1 | 62.5 | 37.7 |

*Table 8.* Quantitative result comparison between Segment Anyword using ground truth text description and GLaMM generated text description.

We conducted an additional experiment using GLAMM-generated captions as textual input for our method, reporting both quantitative results (Table 8) and qualitative results (Fig. 13). In general, since GLAMM-generated text is not always accurate, it can affect the alignment of prompt embeddings in our method. Nevertheless, our approach remains competitive, as it is capable of refining and adapting prompt embeddings—even from noisy or imprecise text—to better match the target object during test-time optimization.

## F.3. Ablation study with Pure Text Prompt

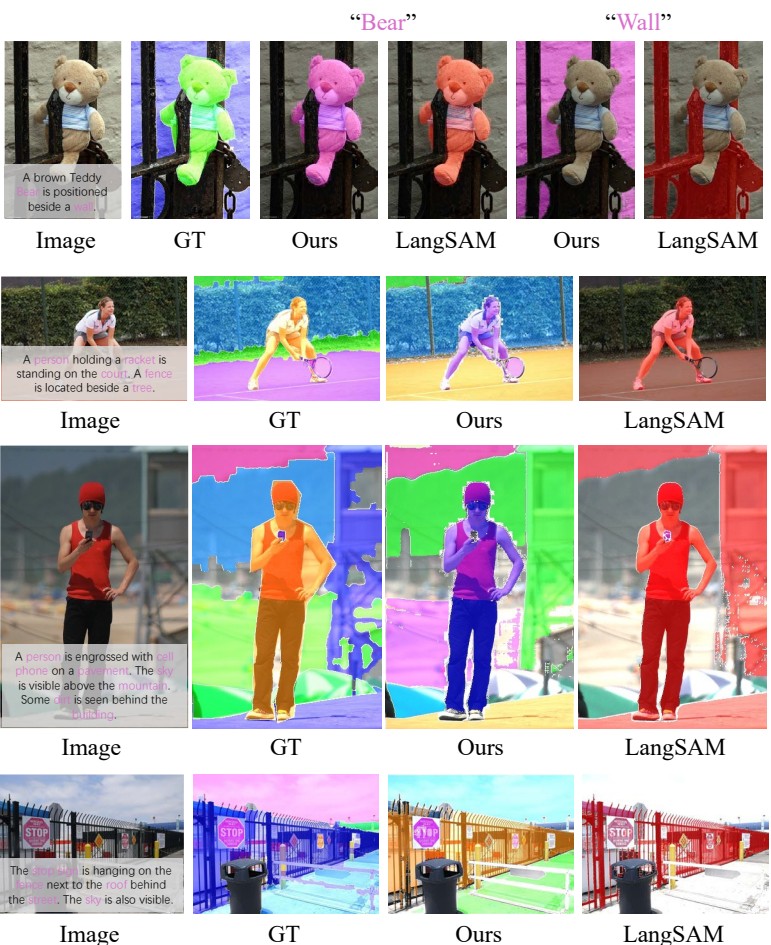

*Figure 14.* Qualitative result comparison between Segment Anyword and Segment Anything Model with text description and prompt(LangSAM).

|  | GranDf Val | |
|---|---|---|
|  | mAP | mIoU |
| Segment Anyword | 31.3 | 67.4 |
| LangSAM | 17.6 | 33.5 |

*Table 9.* Quantitative result comparison between Segment Anyword and Segment Anything Model with text description and prompt(LangSAM).

We presnet additional experiments of how different prompt influence the downstream mask generation. We compare our Segment Anyword mask prompt against pure text input, using official implementation of LanguageSAM[2]. We present both quantitive and qualitative results on GranDf validation set. Results show that our method is very effective on improving mask prompt quality.

---

[2]https://github.com/luca-medeiros/lang-segment-anything

## F.4. Ablation study using llm generated text description

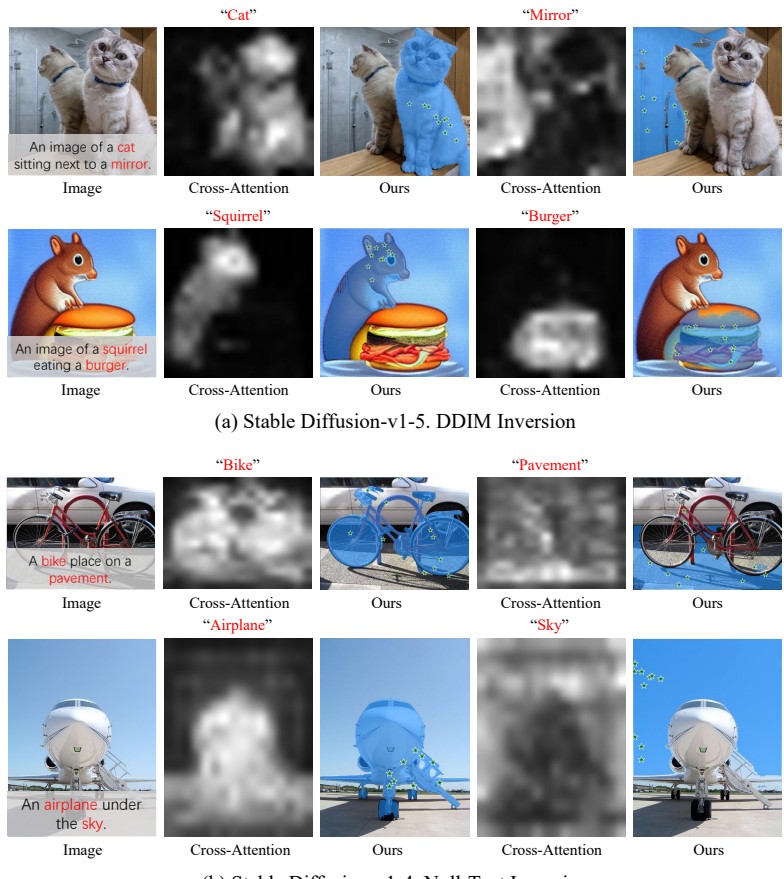

*Figure 15.* Qualitative result comparison between Segment Anyword using different diffusion backbones (Stable Diffusion v1.4 and v1.5) and different inversion algorithms (DDIM inversion and null-text inversion).

MCPL primarily serves as the inversion backbone in our method. However, our approach is not limited to MCPL and can be seamlessly integrated with other inversion or concept-discovery methods. We present qualitative results (Fig.15) using different cross-attention sources, demonstrating that our method is composable with various diffusion models and inversion algorithms. These include replacing LDM with Stable Diffusion(Rombach et al., 2022) versions 1.4 and 1.5, and using alternative inversion techniques such as vanilla DDIM inversion (Song et al., 2021) and Null-Text Inversion (Mokady et al., 2023).

# G. Additional Stress Testing Results

## G.1. Grounded Segmentation with Novel Visual Concepts

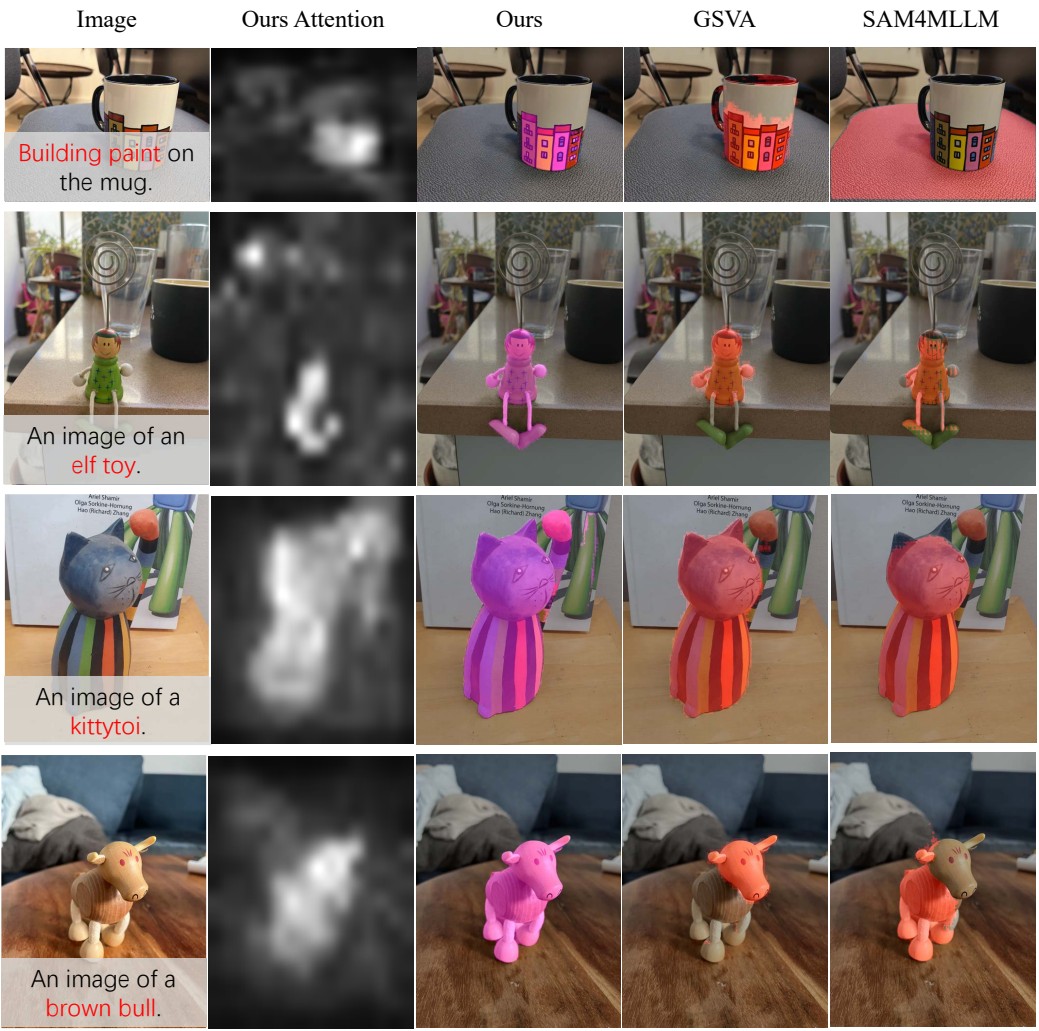

Figure 16. **Stress Testing Results:** We show qualitative results from stress testing study, where the test data involves novel visual concepts.

We present a qualitative comparison of stress testing involving several novel concepts, such as building paint, kittytoi, and brown bull. For GSVA (Xia et al., 2024), we use the LLaMA-7B base weights with LLaVA-Lightning-7B-delta-v1-1 and SAM ViT-H. For SAM4MLLM (Chen et al., 2024), we employ LLaMA-LLaVA-next-8B and EfficientViT-SAM-XL1. While both methods are capable of localization, they struggle to produce accurate masks, particularly in capturing fine details around object parts and boundaries. It is important to note that both GSVA and SAM4MLLM require training or fine-tuning of LLaVA and SAM on the training set, which demands substantial computational resources. In contrast, our method operates purely at test time without accessing the full training set, making it both simple and effective for handling novel concepts.

## G.2. Grounded Segmentation under Noisy Text Description

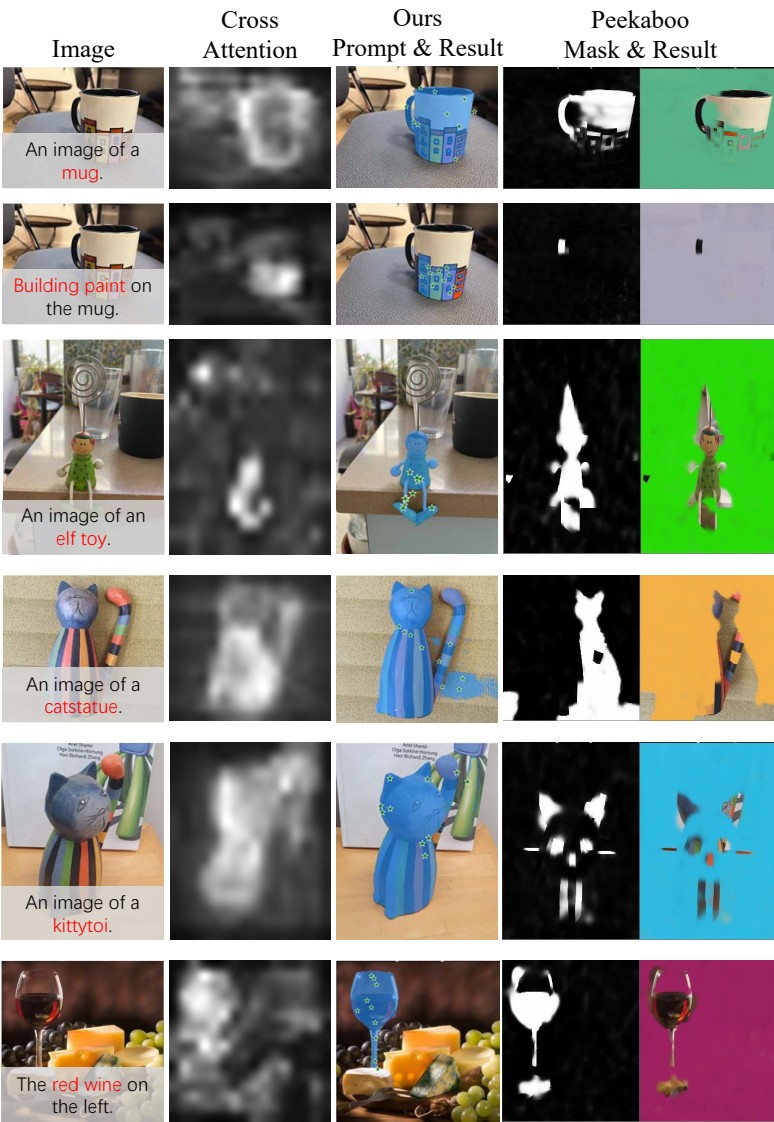

*Figure 17.* **Stress Testing Results:** We show qualitative results from stress testing study, where the test data involves noisy test descriptions.

We acknowledge that noisy inputs, including sentences with incorrect grammar, may be present from customized user input during test time. However, handling noisy text parsing is not the primary focus of our work. Instead, our objective is to generate mask prompts that are robust to such noise, without relying on supervised training. As shown in Fig. 17, our method can effectively handle typos such as "catstatue" and "kittytoi." In extreme cases, word-level cross-attention mask can be directly matched against ground-truth segmentation masks to determine the best ranking and pairing, thereby compensating for parsing inaccuracies.

### G.3. Inference-Time Speed Comparison

| Methods | Average Inference Time /image | Denoising Steps |
|---|---|---|
| Peekaboo (Burgert et al., 2022) | 150s | 300 steps |
| CLIPasRNN (Sun et al., 2024) | 180s | - |
| Segment Anyword | 470s | 1100 steps |
| Segment Anyword$_f$ | 28s | 50 steps |

*Table 10.* Inference time speed comparison between Segment Anyword and other training-free baseline methods including CLIPas-RNN (Sun et al., 2024) and Peekaboo (Burgert et al., 2022).

We acknowledge that a trade-off exists between test-time optimization speed and mask prompt quality. Specifically:

- Previous methods typically require substantial computational resources and manual effort to train or fine-tune large vision-language models on curated training datasets. While effective in controlled settings, these approaches often lack generalization capabilities and are resource-intensive.

- In contrast, our proposed test-time prompt optimization method is more efficient and better suited for real-world open-set scenarios, where test samples may include novel linguistic and visual concepts. In such cases, textual embeddings must be dynamically updated and aligned with the target object during inference. Thus, the trade-off between inference speed and embedding alignment is inherent and cannot be entirely eliminated.

- Nevertheless, we demonstrate that the number of test-time steps can be substantially reduced to improve inference speed. Table 10 presents a comparison of inference time with related training-free baselines, including CLIPasRNN (Sun et al., 2024) and Peekaboo (Burgert et al., 2022). All experiments were conducted on a single NVIDIA A100 40GB GPU. Our method shows significant acceleration—reducing inference time from 470s to 28s by fine-tuning the text encoder on a small number of target-domain samples and decreasing the number of inference steps from 1100 to just 50, with minimal performance degradation. Future work may further enhance speed through engineering optimizations, such as replacing the current backbone with Hyper Stable Diffusion (Ren et al., 2024).

# H. Additional Qualitative Segmentation Results

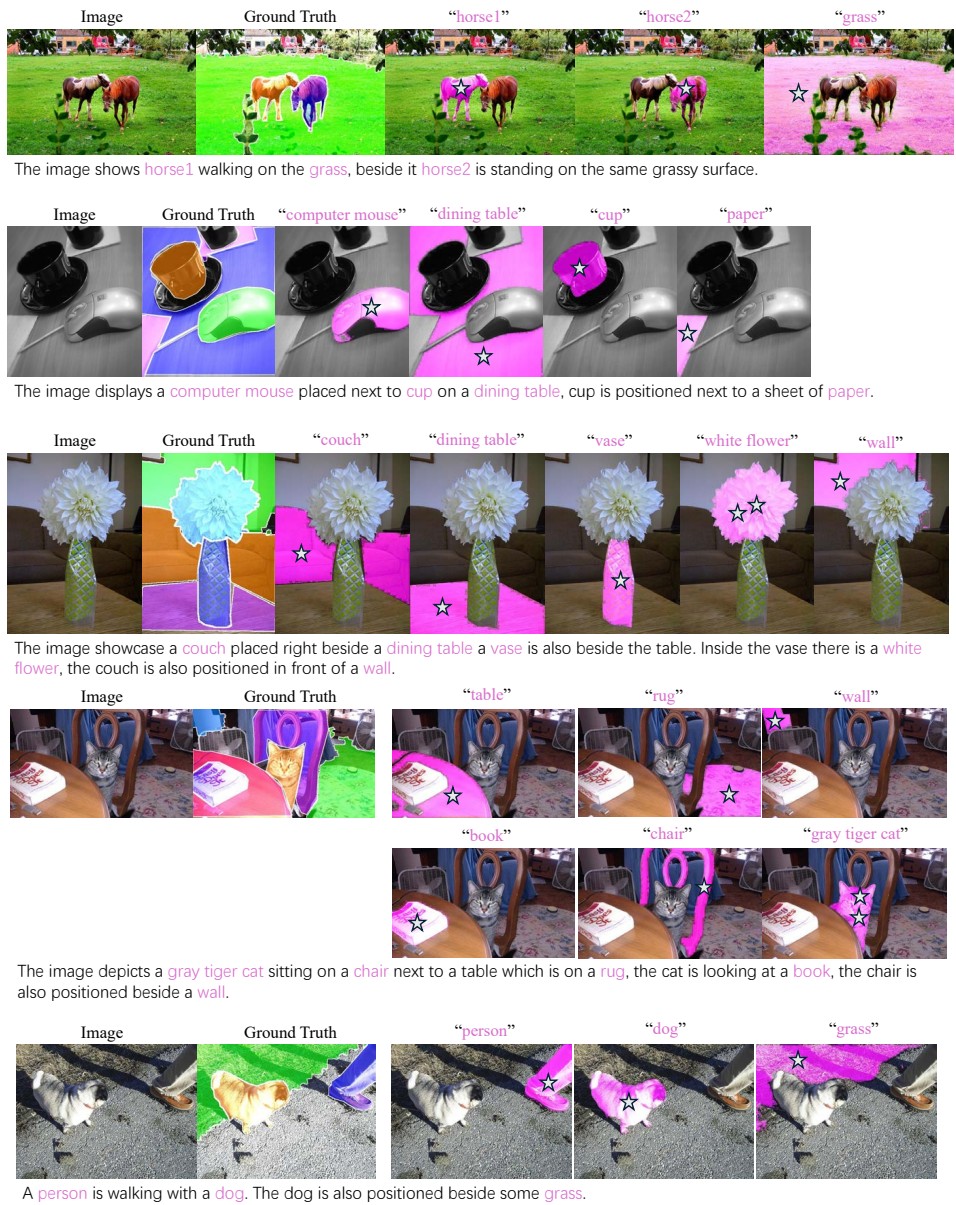

Figure 18. **Additional qualitative open-set language grounded segmentation results (GranDf validation set).** We show that Segment Anyword can achieve accurate open-set language grounded segmentation with well-localized mask prompt, even can generate fine-grained object part-aware masks such as "chair", "vase" and "rug"

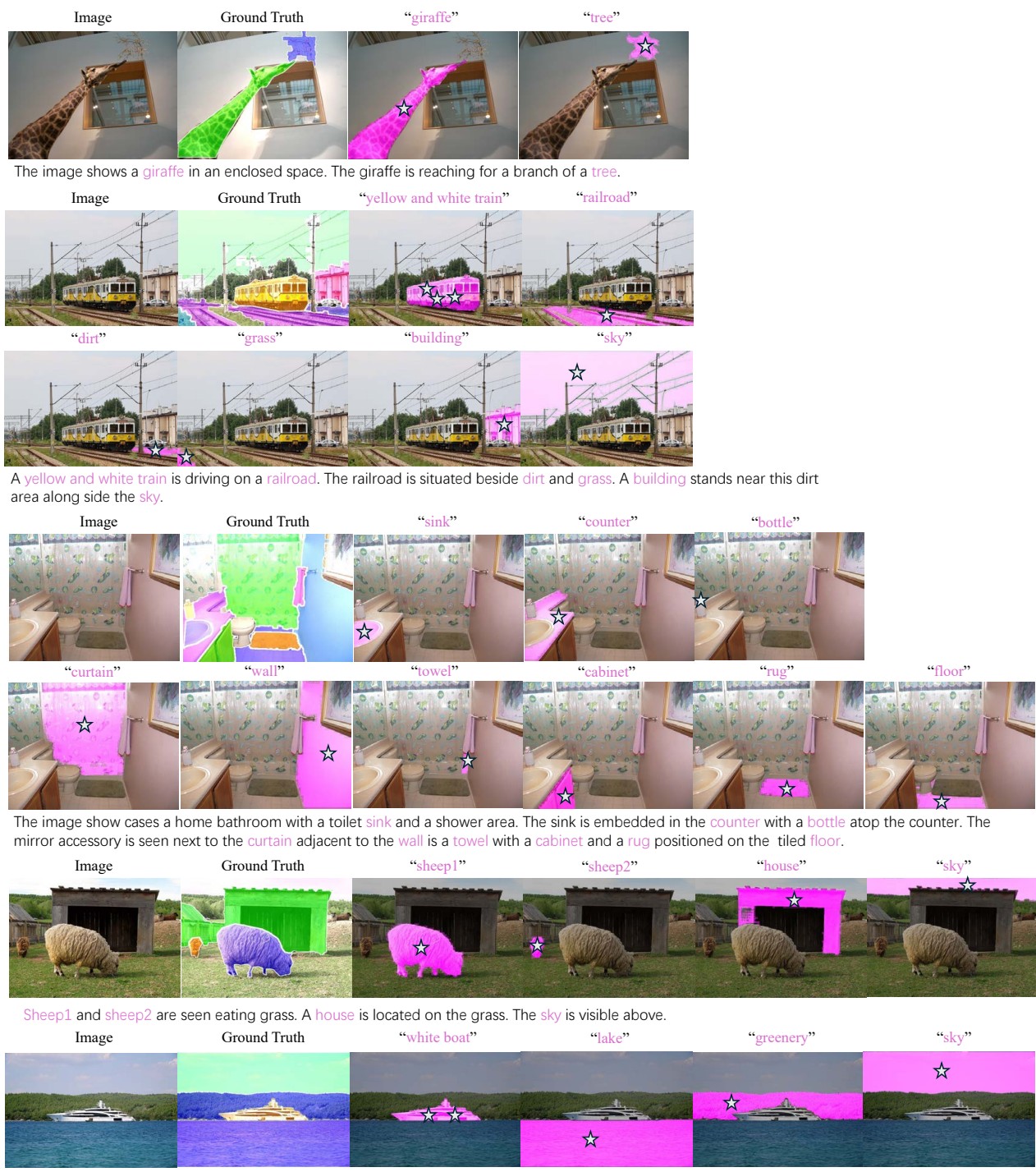

*Figure 19.* **Additional qualitative open-set language grounded segmentation results (GranDf validation set).** We show that Segment Anyword can achieve accurate open-set language grounded segmentation with well-localized mask prompt, even with complex indoor placements "counter" and outdoor infrastructure "railroad".

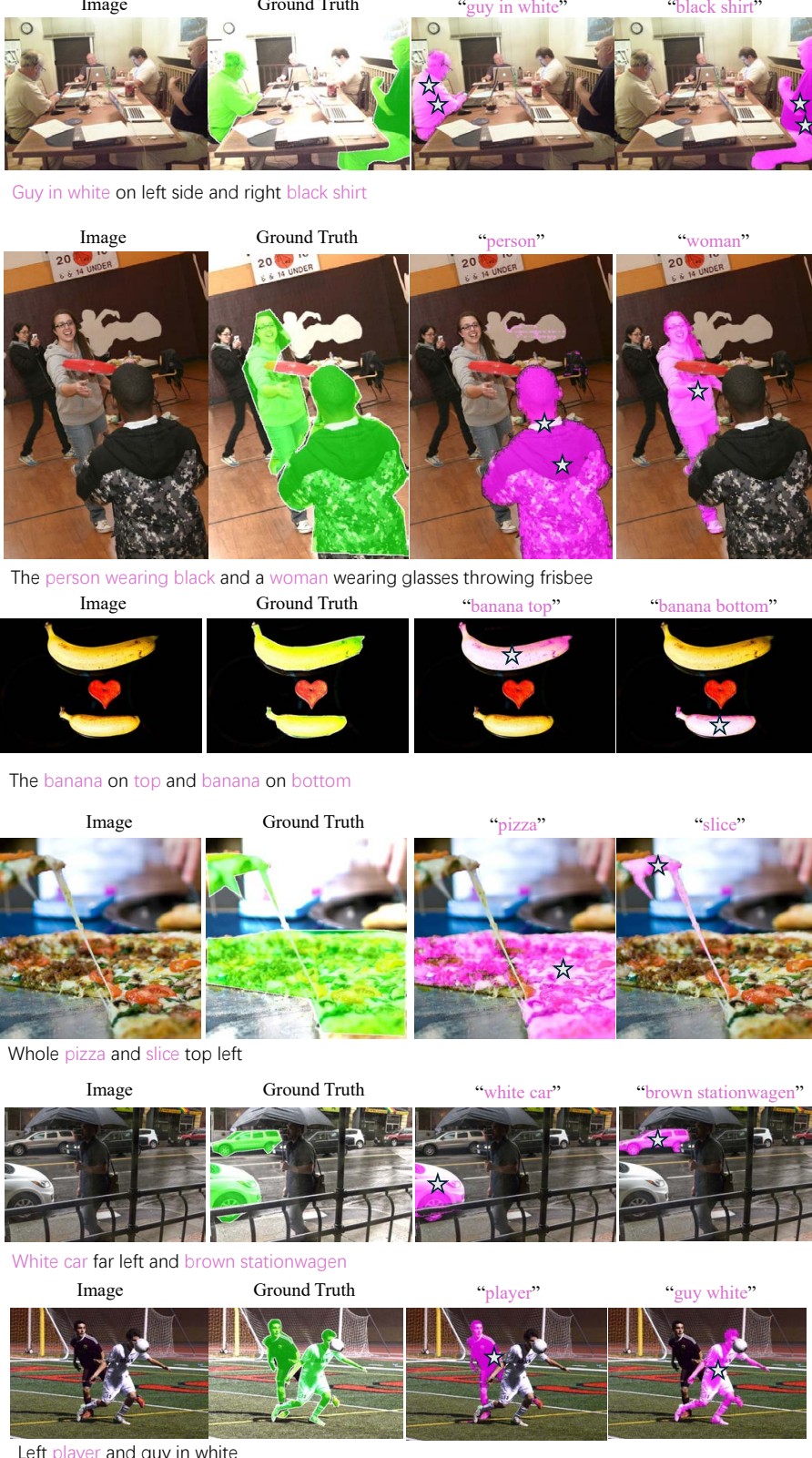

*Figure 20.* **Additional qualitative reference image segmentation results (gRefCOCO validation set).** We show that Segment Anyword can achieve accurate reference image segmentation with well-localized mask prompt. With updated textual embedding, it can further distinguish "slice" from "pizza"

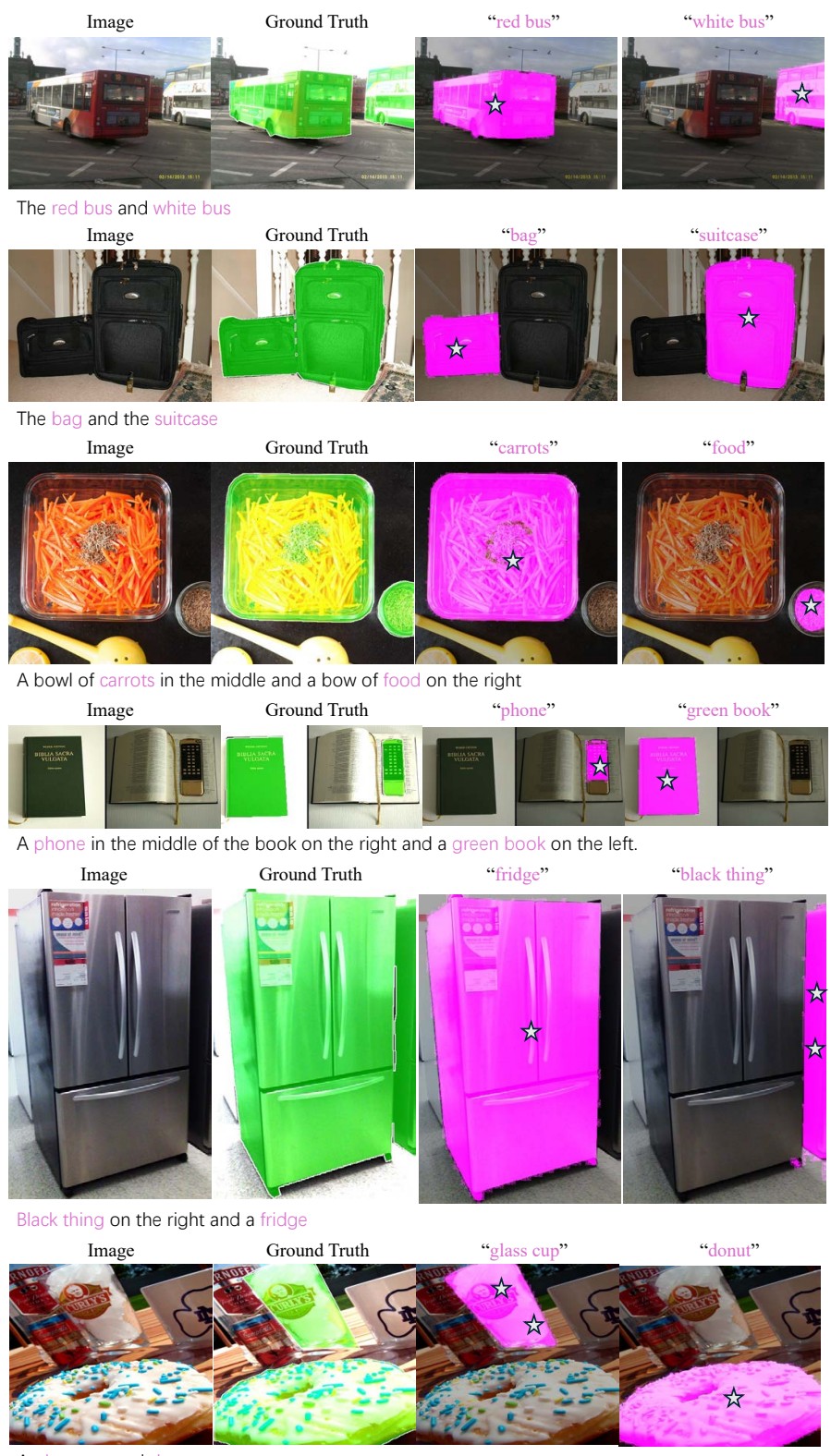

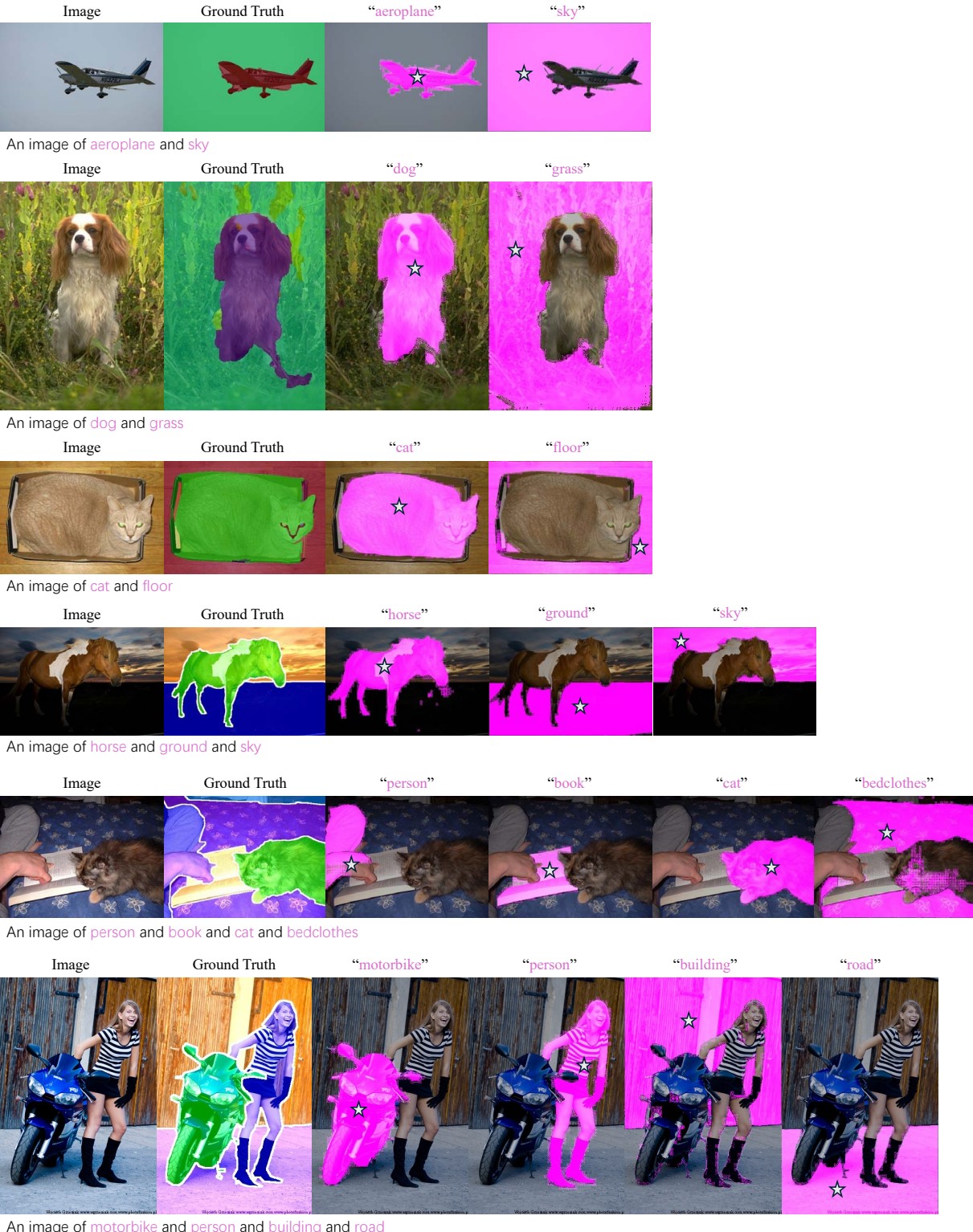

*Figure 22.* **Additional qualitative open-vocabulary semantic segmentation results (Pascal Context 59 validation set).** By concatenating word label into sentences, we show that Segment Anyword can achieve accurate open-vocabulary image segmentation with well-localized mask prompt.

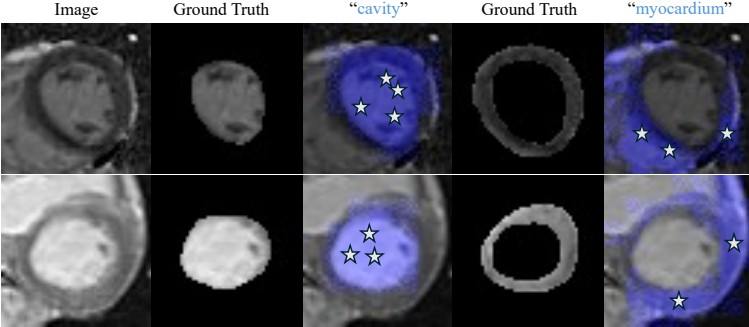

an image of round cavity encircled by circle myocardium

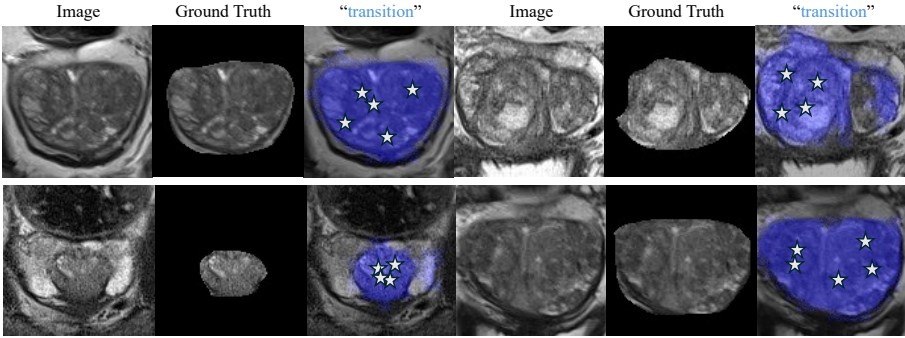

an image of round transition

*Figure 23.* **Exemplar qualitative OOD Medical Image segmentation results.** We show that Segment Anyword has a potential prompt learning and segmentation capability for out-of-distribution medical image segmentation, such as "cavity", "myocardium" and prostate "transation" zone.

# I. Additional Results on Failure Cases

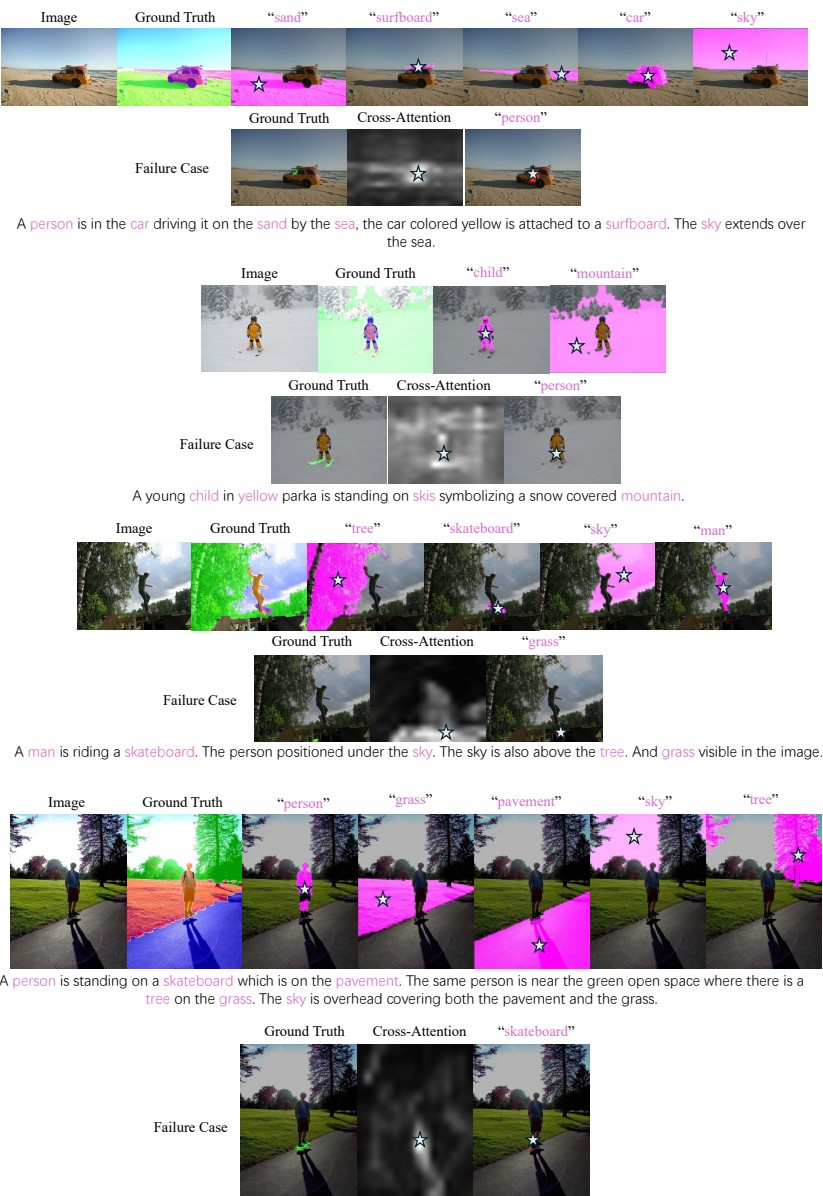

*Figure 24.* **Additional qualitative failure case results.** We acknowledge that Segment Anyword can still encounter failure segmentation, particularly when the target object is small and visual ambigious, such as "skis" and "skateboard". This is mainly due to the cross-attention map resolution is small (16×16), which may lead to a wrong localization of mask prompt back to the orginal image, although the diffusion backbone can capture the visual concept.

