# OpenReview forum: "Segment Anyword: Mask Prompt Inversion for Open-Set Grounded Segmentation"
_ICML.cc/2025/Conference — ICML 2025 poster_

### Official Review · Reviewer_SSnW · 2025-03-11

**Overall Recommendation:** 4

**Summary:**

The authors identify an issue in VLMs / MLLMs of unstable segmentation against variations in textual prompt. They propose a text-to-image diffusion model based test-time optimization technique combined with language guided prompt tuning to solve this issue. Their resulting framework, tagged Segment Anyword, is evaluated across diverse tasks and datasets to show strong performance.

**Claims And Evidence:**

Inadequate evidence on how method solves the motivation problem of how VLMs *struggle with diverse terms* in textual prompt. Provide some simple quantitative evaluation (maybe your own benchmark using the dataset used in Figure 3) to back this claim. This will strengthen the paper. See more info in weaknesses.

**Essential References Not Discussed:**

N/A

**Experimental Designs Or Analyses:**

Yes, good.

**Methods And Evaluation Criteria:**

Yes.

**Other Comments Or Suggestions:**

Consider discussing related work:

* Ranasinghe, K., McKinzie, B., Ravi, S., Yang, Y., Toshev, A., & Shlens, J. (2023). Perceptual Grouping in Contrastive Vision-Language Models. 2023 IEEE/CVF International Conference on Computer Vision (ICCV), 5548-5561.

* Mukhoti, J., Lin, T., Poursaeed, O., Wang, R., Shah, A., Torr, P.H., & Lim, S.N. (2022). Open Vocabulary Semantic Segmentation with Patch Aligned Contrastive Learning. 2023 IEEE/CVF Conference on Computer Vision and Pattern Recognition (CVPR), 19413-19423.

* Burgert, R., Ranasinghe, K., Li, X., & Ryoo, M.S. (2023). Peekaboo: Text to Image Diffusion Models are Zero-Shot Segmentors. CVPR 2023.

**Other Strengths And Weaknesses:**

**Strengths**

1. Interesting motivation analysis
2. Tackles a difficult task of reducing VLM sensitivity to input prompts
3. Through evaluation to show consistent performance improvements across benchmarks.



**Weaknesses**


1. Sec 2.2 / Figure 3 unclear: please provide more details regarding the visualization in the Figure caption.
    * What is the exact model used to generate segmentations here?
    * In plot, is IoU calculated against the ground-truth?
    * What exactly does each dot correspond to? A single image with multiple captions (mean / std along caption dimension)?
    * Can you use a different colour for the second red? You use red for two examples
    * What's inside brackets? I'm assuming it's IoU against GT?
    * "Thus we extend each image associating with additional generated 2-5 mutated expressions (e.g. [ apple pieces]→ [ apple pieces, apple slices, cut up apples])." - how are these generated? Templates? LLM?

2. Accomplishment of motivation.
    * How does Figure 3 plot look if you add your model there?
    * Can you provide table with average std (i.e. calculate std for each image as done currently, and then average across all images) of baseline vs your method?
    * If the problem identified in contributions (1) - *struggle with diverse terms* - is solved by method, the average std (in an experiment like Figure 3) should drop for your method compared to baseline. Is this correct? Discuss this and back this motivation with quantitative results.

3. Inference Speed (time / compute)
    * This method appears much slower compared to other approaches that do not use test-time optimization
    * Please provide a table comparing the inference time with that of baselines
    * Maybe you can compare to slower, similar test-time optimization method like [1] to justify a slow speed
    * "Training-free" claim in abstract / intro - is this fair, since you are training parameters during inference? In fact, you have at table showing the parameters you *train*.

4. Method details missing
    * "we parse the text expression into Noun-Phrases (NP) and Verb-Phrases (VP) and identify each rooted noun subject (root) within the phrase." - how is this done? What algorithms / methods are used? How accurate is this?
    * If baselines are given this additional information (noun - verb phrases), will they perform as well?


[1] Burgert, R., Ranasinghe, K., Li, X., & Ryoo, M.S. (2023). Peekaboo: Text to Image Diffusion Models are Zero-Shot Segmentors. CVPR 2023.

**Questions For Authors:**

N/A

**Relation To Broader Scientific Literature:**

Relevant and useful topic explored.

**Theoretical Claims:**

N/A

---

> ### Author Rebuttal · Authors · 2025-03-31
>
> __Q1__: "__Exact model to generate segmentations__"
>
> __R1__: We use the official implementation and pretrained checkpoint from ReLA. The final segmentation mask is produced by the pixel decoder whose outputs are weighted by a language-supervised object activation map.
>
> __Q2__: "In plot, is IoU calculated against the ground-truth?"
>
> __R2__: Yes.
>
> __Q3__: "__Each dot means?__"
>
> __R3__: Yes, each dot corresponds to a single image paired with multiple descriptions. The mean and standard deviation are calculated along the caption dimension.
>
> __Q4__: "__Use different colour__"
>
> __R4__: We will revise the color coding in the figures.
>
> __Q5__: "__IoU inside brackets?__"
>
> __R5__: Yes it is IoU against GT.
>
> __Q6__: "__how to generate mutated expressions?__"
>
> __R6__: We generate the mutated expressions by querying ChatGPT4o, with a template "_Genrate a list of __n__ synonyms of the noun phrases in the following [sentence] and output the list separated by '&'_", where __n__ is in randint(2, 5) and _[sentence]_ is set to text referring expression.
>
> __Q7__:"__Figure 3 plot add your model?__"
>
> __R7__: Please find link3 and link4 in our reply R10 to reviewer 4Tbz.
>
> __Q8__: "__Table with average std__"
>
> __R8__: We provide the table with average std below:
>
> | STD Avergage Comparison |          |
> |-------------------------|----------|
> |                         | RefCOCO+ |
> | ETRIS                   |   10.958 |
> | Segment Anyword         |    8.021 |
> |                         |          |
> |                         | gRefCOCO |
> | RELA                    |   13.217 |
> | Segment Anyword         |    4.365 |
>
> __Q9__: "__the average std should drop for your method?__"
>
> __R9__: That's correct. We focus on test-time prompt embedding alignment. We show that out method is simple but very effective to reduce the sensitivity of input variance.
>
> __Q10__: "__Slow inference speed__"
>
> __R10__: Thank you for your feedback. Please refer our reply to reviewer jmV5.
>
> __Q11__: "__inference time comparison__"
>
> __R11__: We present a Table comparing the inference time with related baselines, including CLIPasRNN, Peekaboo.
>
> |                   | Average Speed Per Image | Inference Steps Per Image |
> |-------------------|-------------------------|---------------------------|
> | Segment Anyword   | 470s                    | 1100steps                 |
> | Segment Anyword_f | 28s                     | 50steps                   |
> | Peekaboo          | 150s                    | 300steps                  |
> | CLIPasRNN         | 180s                    | -                         |
>
> All implementations were evaluated on a single NVIDIA A100 40GB GPU. As previously discussed, test-time optimization methods inherently involve a trade-off between inference speed and mask quality. Nevertheless, we demonstrate that our method can be significantly accelerated—reducing inference time from 470s to 28s—by fine-tuning the text encoder on a small number of target-domain samples and decreasing the number of inference steps from 1100 to just 50, with minimal performance degradation.
>
> __Q12__: "__Training-free overclaim?__"
>
> __R12__: Please refer our reply R1, R2 and R3 to reviwer 4Tbz.
>
> __Q13__: "__Concerns on expression parsing__"
>
> __R13__: Please refer our reply R4 and R5 to reviewer ixNi
>
> __Q14__: "__baseline + additional info__"
>
> __R14__: We observe that certain baseline methods, such as GLamm, LISA, and OMG-LLaVA, already incorporate similar linguistic information into their input pipelines. For instance, by integrating complete noun-phrase during feature fusion, or translating a special [SEG] token with context into segmentation masks. These models achieve impressive results but require significant training efforts.
>
> Other baseline methods focused on test-time optimization, such as CLIPasRNN, OVDiff, and Peekaboo, could theoretically leverage this linguistic information as auxiliary input; however, empirical observations suggest that their performance remains suboptimal under these conditions.
>
> In contrast, our proposed method explicitly formalizes this linguistic knowledge as prompt regularization. This strategy enables robust mining of noise-tolerant mask prompts and consequently yields refined and higher-quality segmentation masks, even without extensive training or configuration overhead.
>
> __Q15__: "__Discussing related works.__"
>
> __R15__: We thank the reviewer for highlighting these important works, which also aim to leverage intermediate features from large vision-language models (VLMs) for downstream segmentation tasks.
>
> Both CLIPpy and PACL assign labels by contrasting image patch embeddings with object text label embeddings. However, these methods still require training configurations.
>
> Peekaboo is closely related to ours, as both aim to learn visual concepts using off-the-shelf diffusion models. However, Peekaboo heavily relies on alpha map initialization and is sensitive to its quality.
>
> We will add this discussion to our manuscript.

---

### Official Review · Reviewer_4TbZ · 2025-03-12

**Overall Recommendation:** 2

**Summary:**

This paper introduces Segment Anyword, a training-free framework for open-set language-grounded image segmentation. It leverages token-level cross-attention maps from a frozen diffusion model to generate mask prompts, which are then refined by the Segment Anything Model (SAM) for accurate segmentation. To address the lack of coherence and consistency in initial prompts, the authors propose Linguistic-Guided Visual Prompt Regularization, which incorporates syntactic and dependency structures to improve prompt quality. The method shows significant performance gains and strong generalization across diverse datasets without the need for extensive training or fine-tuning.

**Claims And Evidence:**

Overall, the paper presents its claims with clear and convincing evidence.

The authors support their critique of existing open-set segmentation methods—namely, their reliance on extensive training and difficulty in achieving consistent object segmentation across diverse textual expressions—through both quantitative and qualitative analysis, particularly in Figure 3.

To address the issue of incoherent and inconsistent prompt quality, the proposed linguistic-guided visual prompt regularization enhances alignment between text expressions and visual masks. The effectiveness of this module is demonstrated both qualitatively (Figure 6) and quantitatively (Table 6).

The claim that the method is computationally lightweight compared to existing approaches is substantiated in Table 1, where the number of trainable parameters is significantly lower than that of most prior methods.

However, the repeated emphasis on being "training-free" warrants scrutiny. In practice, the method updates textual embeddings for up to 1100 steps during inference, which arguably constitutes test-time adaptation or lightweight training. Additionally, the use of LoRA to fine-tune the BERT encoder further complicates the notion of being entirely training-free. While not the central contribution, this aspect should be more carefully framed to avoid overstating the claim.

Finally, although the method effectively provides initial localization through cross-attention maps, much of the segmentation accuracy relies heavily on the use of SAM as a post-processing module. This dependency raises questions about how much of the final segmentation performance can be attributed solely to the proposed method itself.

**Essential References Not Discussed:**

Prompt learning papers, beginning with "CoOp: Conditional Prompt Learning for Vision-Language Models" as the seminal work, should be addressed, followed by its successors. CoOp’s experimental design is empirically sound, introducing prompt learning to adapt vision-language models like CLIP for few-shot tasks, efficiently generating masks and outperforming baselines like zero-shot CLIP across datasets such as ImageNet; however, it struggles with unseen classes. Follow-up works like "CoCoOp" (CVPR 2022) enhance this by adding input-conditional adaptability for open-world segmentation of diverse text expressions, maintaining soundness with robust testing but introducing computational complexity.

**Experimental Designs Or Analyses:**

Most of the experiments are based on empirical results and are considered sound.

**Methods And Evaluation Criteria:**

The proposed method is well-aligned with the problem of open-set language-grounded segmentation and effectively addresses its key challenges. The evaluations are conducted on appropriate benchmark datasets using standard metrics, and the results, along with ablation studies, support the method’s validity.

**Other Comments Or Suggestions:**

entirely : off-shelf → off-the-shelf?
L296 : eight → seven?

**Other Strengths And Weaknesses:**

The core framework claimed in this paper heavily borrows from the assertions in "An Image is Worth Multiple Words: Discovering Object-Level Concepts using Multi-Concept Prompt Learning," but it lacks sufficient evidence to explain what differentiates the proposed method. It directly adopts the approach of averaging cross-attention maps and merely refines the masks using SAM afterward. Given that this is a training-free method not utilized for learning, I believe a more significant approach is needed.

Relatedly, I think the ablation study for the proposed method is insufficient. There’s little detail on how SAM is applied or what design choices are made in the subsequent regularization step, where deeper design exploration and ablation studies seem necessary.

Overall, the explanations for figures and tables are lacking. In Figure 3, the intent is somewhat clear, but it’s unclear what the green, blue, and red colors represent. Similarly, in Table 6, terms like PL, R1, and R2 are presented without any explanation of their meaning.

**Questions For Authors:**

As mentioned earlier, I have doubts about the novelty of this paper. Most of the ideas are derived from existing methods, and reinforcing the noun part of text descriptions with adjectives or adverbs isn’t particularly a new idea either. This concern needs to be addressed.

**Relation To Broader Scientific Literature:**

The proposed method focuses on generating masks more efficiently compared to existing methods. Being able to effectively perform segmentation for various text expressions in an open-world scenario can be highly impactful.

**Theoretical Claims:**

The paper does not present formal theoretical claims; it is primarily empirical, focusing on experimental results and practical effectiveness.

---

> ### Author Rebuttal · Authors · 2025-03-31
>
> We sincerely thank the reviewer for their detailed feedback. We are glad that the reviewer found our paper to present clear claims with evidence, supported critiques. And our method is computationally lightweight.
>
> __Q1__: "__training-free warrants scrutiny__"
>
> __R1__: The term “training-free” refers to the fact that our method does not require access to or processing of the training dataset. In contrast to prior approaches that demand significant computational resources to train or fine-tune on curated datasets, our method avoids such data-dependent training procedures.
>
> Instead, we perform at test-time, which is more suitable for real-world, open-set scenarios—where test samples often involve novel concepts that has not been seen during training. In such cases, updating prompt embeddings during inference is necessary to align them with the target objects effectively.
>
> To avoid confusion, we will revise our terminology throughout the manuscript, replacing “training-free” with “test-time optimization.”
>
> __Q2__: "__fine-tune the BERT encoder increase model complexity.__"
>
> __R2__: We demonstrate that it is possible to reduce the number of diffusion steps during test time textual embedding update from 1100 to as few as 50—by adapting the text encoder using a small number of samples, without accessing the full training dataset. This lightweight adaptation, achieved via LoRA, is intended purely for accelerating test-time optimization, not for conventional training or fine-tuning on a full dataset.
>
> Importantly, this step does not involve large-scale training or supervision and serves as a practical means to align the text encoder with the target domain efficiently. We emphasize that this adaptation remains consistent with our goal of avoiding full model training and supports fast, sample-efficient test-time optimization.
>
> We will add this discussion to our manuscript.
>
> __Q3__: "__avoid overstating claim.__"
>
> __R3__: We are sorry for the confusion terms and we will replace with "test-time optimization" for clarification.
>
> __Q4__: "__Proposed method's impact on performance.__"
>
> __R4__: We thank the reviwer for raising this issue. We presnet additional experiments of how different prompt influence the downstream mask generation. We compare our Segment Anyword's mask prompt against pure text input, using official implementation of LanguageSAM. We present both quantitive and qualitative results [link1](https://anonymous.4open.science/api/repo/ICML-Submission3702-Rebuttal/file/Imgs/Ours%20vs%20LangSAM.pdf) on GranDf validation set. Results show that our method is very effective on improving mask prompt quality.
>
> |                 | GranDf Val |      |
> |-----------------|------------|------|
> |                 | mAP        | mIoU |
> | Segment Anyword |       31.3 | 67.4 |
> | LangSAM         |       17.6 | 33.5 |
>
> We will add this additional experiment to our manuscript.
>
> __Q5__: "__Discuss with CoOp and CoCoOp__"
>
> __R5__: We thank the reviewer for bringing up CoOp and CoCoOp, two important works in visual prompt learning. We agree with reviewer's comments and will include the discussion in our manuscript.
>
> __Q6__: "__difference from MCPL__"
>
> __R6__: MCPL serves primarily as the inversion backbone in our method. However, our approach is not limited to MCPL and can be seamlessly integrated with other inversion or concept-discovery methods as well. We present a qualitative results with different cross-attention source, showing that our method is composible with various diffusion models and inversion algorithms [link2](https://anonymous.4open.science/api/repo/ICML-Submission3702-Rebuttal/file/Imgs/Ours%20with%20different%20cross%20attention%20source.pdf). We will add this discussion to our manuscript.
>
> __Q7__: "__green, blue, and red represent?__"
>
> __R7__: These colors illustrates easy/medium/hard samples for achieve accurate and stable segmentation, categorized by its IoU mean and std.
>
> __Q8__: "__PL, R1, and R2 represent?__"
>
> __R8__: PL stands for prompt learning. R1 and R2 are two linguistic guided prompt regularizations.
>
> We will revise the figure and table captions to enhance clarity and avoid ambiguity.
>
> __Q9__: __Typos__
>
> __R9__: We will revise our typos throughout the paper.
>
> __Q10__: "__novelty of this paper__"
>
> __R10__: As demonstrated through comprehensive experiments and additional results provided in [link3](https://anonymous.4open.science/api/repo/ICML-Submission3702-Rebuttal/file/Imgs/gRefCOCO_ext_Scatter_comparison.pdf) and [link4](https://anonymous.4open.science/api/repo/ICML-Submission3702-Rebuttal/file/Imgs/RefCOCO_TestB_Scatter_comparison.pdf), our method effectively improves both accuracy and stability by leveraging an off-the-shelf diffusion model to extract mask prompts via an inversion process. This approach is modular and plugable, offering several advantages—including the ability to handle novel visual and linguistic concepts—without requiring additional supervision or complex training procedures.

---

### Official Review · Reviewer_ixNi · 2025-03-14

**Overall Recommendation:** 2

**Summary:**

This work proposes Segment Anyword, an approach for language-guided open-set segmentation. It uses a diffusion model to create initial correspondence between words in the text prompt and points in the image, refines the point prompts based on linguistic analysis, and prompts SAM to generate the final segmentation masks. Empirical results on tasks like referring image segmentation show promising results.

**Claims And Evidence:**

- Table 1 introduces some attributes. To the reviewer, "Word-Grounding" and "Novel-Concept" are not clearly defined. It is also unclear why the most recent models like GSVA and SAM4MLLM do not satisfy these conditions.

- The motivation study is limited to ReLA, a previous model not equipped with large language models. The conclusion of this study may not hold true for more recent MLLM-based models.

**Essential References Not Discussed:**

No concerns.

**Experimental Designs Or Analyses:**

- The experiment on "open-set grounded segmentation" is very misleading. In the original work, GLaMM, GranDf is proposed for the "grounded conversation generation" task, where a model is required to generate a detailed description of the image and ground the noun phrases within the description. GLaMM does not mention the task as "open-set grounded segmentation." Furthermore, as detailed in Appendix B.2, this work "only focus on segmentation capability evaluation by using ground truth text expression as segmentation reference." The comparison in Table 2 is not fair and the performance of Segment Anyword cannot be considered as high, given that the text descriptions are from the ground truth.

**Methods And Evaluation Criteria:**

- The method heavily relies on an external model (a fine-tuned Vicuna, as indicated in Appendix B.1) and a tagging method to parse the text prompt, whose parsing accuracy is not validated. In reference image segmentation, some text expressions are written by human annotators with incorrect grammar structures. How to correctly understand these expressions when the parsing is noisy?

**Other Comments Or Suggestions:**

No other comments.

**Other Strengths And Weaknesses:**

No other concerns.

**Questions For Authors:**

- In the test-time cross-attention collection step, which layer(s) of the denoising UNet are considered when collecting the cross-attention values?

**Relation To Broader Scientific Literature:**

This work introduces a new framework for open-set segmentation with language prompts without leveraging extensive supervision. However, the experiment designs have several flaws and even misleading results, which should be fixed before publication. The proposed method is also a bit too complicated and relies on several external models (diffusion model, parsing LLM and SAM), limiting its applicability in a wider range of tasks.

**Theoretical Claims:**

This work does not include theoretical claims.

---

> ### Author Rebuttal · Authors · 2025-03-31
>
> We thank the reviewer for their thoughtful questions and detailed feedback, which have significantly helped improve the quality of our manuscript. Below, we address the reviewer’s concerns point by point.
>
> __Q1__: "__Definition on "Word-Grounding" and "Novel-Concept"__"
>
> __R1__: We refer "word-grounding" to the model’s ability to associate or align words from a sentence prompt with specific object regions in an image.
> We refer "novel-concept" to that the model can recognize or handle new concepts it has not encountered during training.
>
> We will make amendements on Table caption.
>
> __Q2__: "__Additional motivation study.__"
>
> __R2__: We thank the reviewer for raising this question. We present additional motivational study with another state-of-the-art model ETRIS (ICCV23) ([link1](https://anonymous.4open.science/api/repo/ICML-Submission3702-Rebuttal/file/Imgs/ETRIS-RefCOCO+_TestB-Scatter.pdf)), which focus on aligning representation from pre-trained visual and language encoder by adding intermediate fine-tuned adapters. The result also validates our previous findings that without test-time alignment, current open-set segmentation models suffer from input variance, resulting in unstable segmentation results.
>
> We will extend this discussion to motivation section in our manuscript.
>
> __Q3__: "__why baseline methods failed with certain conditions.__"
>
> __R3__: Both GSVA and SAM4MLLM generate multiple object masks within a single binary segmentation map, but these masks are not explicitly associated with specific word indices. Additionally, neither model is capable of handling novel concepts, as both require training or fine-tuning on the target dataset. This limits their ability to generalize to unseen concepts during test time.
>
> We will add this extended discussion to our manuscript.
>
> __Q4__: "__Text parsing relies on external model__"
>
> __R4__: The primary focus of the proposed Segment Anyword is to improve the quality of automatic mask prompts without relying on complex training configurations (Page 2 Contribution 2). The external language model is merely used to parse and index object-related words for cross-attention map retrieval. Importantly, the use of a fine-tuned LLM (e.g., Vicuna) is not required—our method is compatible with state-of-the-art language models such as GPT-4o, which can provide strong reasoning capabilities out of the box.
>
> Additionally, standard NLP libraries such as NLTK and SpaCy can be used for text pre-processing as well. These tools are widely adopted in prior work, known to be fast and reliable.
>
> We will add this discussion to our manuscript.
>
> __Q5__: "__Text could be noisy__"
>
> __R5__: We acknowledge that noisy inputs, including sentences with incorrect grammar, may occur in the test dataset. However, addressing noisy text parsing is not the focus of our work. Instead, our goal is to generate mask prompts that are robust to such noise, without relying on supervised training. We show an example that our method can still handle typos such as "_catstatue_" and "_kittytoi_" in [link2](https://anonymous.4open.science/api/repo/ICML-Submission3702-Rebuttal/file/Imgs/Ours%20vs%20Peekaboo.pdf)
>
> In extreme cases, each word’s cross-attention mask can be matched directly against the ground-truth segmentation masks to determine the best ranking and pairing, thereby compensating for parsing inaccuracies.
>
> We will add these additional results.
>
> __Q6__: "__Fairness on GranDf comparison__"
>
> __R6__: We thank the reviewer for raising this issue. To address it, we conducted an additional experiment using GLamm-generated captions as the textual input for our method, we report both quantitative and qualitative results [link3](https://anonymous.4open.science/api/repo/ICML-3702-Rebuttal2/file/Ours%20GT%20Text%20vs%20GLamm%20Text.pdf)
>
> |                   |      | GranDf Val |        |
> |-------------------|------|------------|--------|
> |                   | AP50 | mIoU       | Recall |
> | Segment Anyword_f |      |            |        |
> | w/ GT Text        | 30.2 |       65.9 |   42.4 |
> | w/ GLamm Text     | 27.1 |       62.5 |   37.7 |
>
> In general, since GLamm-generated text is not always accurate, it can affect the alignment of prompt embeddings in our method. Nevertheless, our approach remains competitive, as it is capable of refining and adapting the prompt embeddings—even from noisy or imprecise text—to better match the target object during test-time optimization.
>
> We will update the experiment and result.
>
> __Q7__: "__In the test-time cross-attention collection step, which layer(s) of the denoising UNet are considered when collecting the cross-attention values?__"
>
> __R7__: We use the cross-attention maps at the 16×16 resolution, averaged across all denoising time steps to obtain the final cross-attention. This setup is inherited from prior works such as MCPL and Prompt2Prompt to ensure a fair and consistent implementation for textual embedding update. We will add this to our implementation section.

---

> > ### Comment · Reviewer_ixNi · 2025-04-04
> >
> > The authors' response is greatly appreciated. However, some previous concerns remain.
> >
> > - The claim that previous methods like GSVA and SAM4MLLM cannot handle novel concepts needs to be verified with qualitative or quantitative results.
> >
> > - Changing the text parsing model to GPT-4o or SpaCy does not address the concern. Again, the parsing accuracy or the final performance of such alternatives should be validated.
> >
> > - For the comparison on GranDf, after switching captions to GLaMM-generated ones, the performance drops below GLaMM and OMG-LLaVAf shown in Table 2.
> >
> > In addition, I think other reviewers' concerns on training (or test-time optimization) complexity and inference efficiency are valid. Given the presence of these concerns, I keep my rating as "weak reject."

---

> > > ### Author Response · Authors · 2025-04-07
> > >
> > > We sincerely thank the reviewer for the additional questions, which have significantly contributed to improving the quality of our manuscript.
> > >
> > > __Q8__: "__Verify claim__"
> > >
> > > __R8__: We present a qualitative comparison involving several novel concepts, such as *building paint*, *kittytoi*, and *brown bull* in [link4](https://anonymous.4open.science/api/repo/ICML-3702-Rebuttal3/file/novel%20concept%20comparison.pdf). For GSVA, we use the LLaMA-7B base weights with __LLaVA-Lightning-7B-delta-v1-1__ and __SAM ViT-H__; For SAM4MLLM, we use LLaMA-LLaVA-next-8b and __efficientvit-sam-xl1__.
> > >
> > > Both methods can localize but struggle to produce accurate masks, lacking detail around parts and boundaries. Note that both GSVA and SAM4MLLM require training or fine-tuning of LLaVA and SAM on the training set, which involves considerable computational resources.
> > >
> > > In contrast, our method operates purely at test time without accessing the full training set—making it both simple and effective for novel concepts.
> > >
> > > To avoid further ambiguity, we will revise Table 1 from "*Novel Concept*" to "*Localize Novel Concept*", and update the classification accordingly.
> > >
> > > __Q9__: "__Parsing Ablation__"
> > >
> > > __R9__: We thank the reviewer for suggesting this important ablation setting. Due to time constraints, we conducted the study using 100 randomly selected image-text pairs from RefCOCO.
> > >
> > > For the SpaCy, we used the en_core_web_trf pipeline based on RoBERTa. We filtered tokens with pos tag "NOUN" and "ADJ", and indexed the token with the nsubj dependency label as the referred object.
> > >
> > > For GPT-4o and Vicuna-7B, we used the following prompt:
> > >
> > > Prompt:
> > >
> > > *As a NLP expert, you will be provided a caption describing an image. Please do pos tag the caption and identify the only one referred subject object and all adjective attributes. Your response should be in the format of "[(attribute1, attribute2, attribute3, ...), object1]"*
> > >
> > > *Conditions:*
> > >
> > > *(1) If the attribute is long, short it by picking one original word.*
> > >
> > > *(2) Please include one original word possessive source into the attributes for the subject.*
> > >
> > > Below, we present the final segmentation results. The corresponding parsing outputs have also been uploaded at [anonymous link5](https://anonymous.4open.science/api/repo/ICML-3702-Rebuttal4/file/refCOCO_testA_Spacy.json) and [anonymous link6](https://anonymous.4open.science/api/repo/ICML-3702-Rebuttal4/file/refCOCO_testA_GPT4o.json). While spaCy offers fast, offline parsing speed, it often misses key adjectives such as color terms like "white". In contrast, GPT-4o provides the most accurate parsing results, capturing fine-grained attributes more reliably.
> > >
> > > | Parsing Ablation | mIoU |
> > > |------------------|:----:|
> > > | GPT4o            | 68.2 |
> > > | spaCy(RoBERTa)   | 46.9 |
> > > | Vicuna-7B        | 59.7 |
> > >
> > > Our empirical recommendation for users is as follows: for large-scale processing where speed is a priority, static NLP libraries such as spaCy are more suitable due to their efficiency. However, for more detailed interactions involving concept learning and prompt refinement, advanced language models like GPT-4o are preferred for their superior reasoning and parsing capabilities. We will include this discussion in the revised manuscript.
> > >
> > >
> > > __Q10__: "__Performance drop using GLamm generated text__"
> > >
> > > __R10__: The performance drop is expected, as the text input shifts from human-annotated ground truth to generated grounded conversations. However, we observe that the performance remains competitive even with GLamm-generated text, which is outperforming or matching several baseline methods that require training or fine-tuning on the entire training dataset.
> > >
> > > __Q11__: "__Other reviewer concern on inference complexity and efficiency__"
> > >
> > > __R11__: The initial concern raised by jmV5 relates to the inference cost of using a diffusion model. We acknowledged that there is a trade-off between the step numbers and mask prompt quality. While this trade-off cannot be entirely eliminated, we pointed out that using HyperDiffusion can possible reduce the denoising steps as future engineering work.
> > >
> > > Additionally, we addressed a related concern raised by reviewer SSnW, leading to raised score from 3 to 4, by comparing our method to other test-time optimization baselines such as Peekaboo and CLIPasRNN. As in SSnW R11 and in [link5](https://anonymous.4open.science/api/repo/ICML-Submission3702-Rebuttal/file/Imgs/Ours%20vs%20Peekaboo.pdf))
> > > , our method achieves stable and accurate results within a reasonable time, without additional input dependency such as mask initialization or object prototype.

---

### Official Review · Reviewer_jmV5 · 2025-03-19

**Overall Recommendation:** 4

**Summary:**

This paper introduces Segment Anyword, a novel framework for open-set grounded segmentation. The key idea is to invert the mask prompting process by leveraging a pre-trained diffusion-based text-to-image model (e.g., Stable Diffusion) to generate high-quality, grounded segmentation masks. The method achieves competitive performance on both in-distribution and out-of-distribution datasets.

**Claims And Evidence:**

The paper claims that Segment Anyword works effectively on both in-distribution and out-of-distribution datasets. This is supported by quantitative results on multiple benchmarks (GranDf, gRefCOCO, PC59, synthetic/medical datasets), showing strong performance, particularly in handling novel categories without retraining.

The paper introduces positive adjective binding and negative mutual-exclusive binding as key components, which are well-supported by clear and convincing evidence in Table 6. The ablation results directly isolate and quantify their impact on segmentation performance, confirming their utility.

**Essential References Not Discussed:**

No

**Experimental Designs Or Analyses:**

The experimental designs are sound

**Methods And Evaluation Criteria:**

Yes

**Other Comments Or Suggestions:**

See weakness

**Other Strengths And Weaknesses:**

Strengths:

The paper introduces a novel approach by inverting the typical mask prompting paradigm. Instead of designing prompts manually, it optimizes them at test-time, leveraging diffusion models' cross-attention layers for precise mask generation.

A significant advantage is that the method requires no additional training of the diffusion model or SAM, making it resource-efficient and appealing for real-world deployment.

The approach demonstrates robust results on both in-distribution and out-of-distribution datasets, confirming its effectiveness in open-set scenarios.

Weakness:

While the method smartly avoids retraining large models, it heavily depends on test-time optimization, involving multiple gradient steps to optimize textual prompts and collect cross-attention maps from a diffusion model. This inference-time overhead can be significant, especially considering the use of diffusion models that already involve multiple denoising steps.

**Questions For Authors:**

NA

**Relation To Broader Scientific Literature:**

Yes

**Theoretical Claims:**

Yes, there is no proof for theoretical claim.

---

> ### Author Rebuttal · Authors · 2025-03-31
>
> We sincerely thank the reviewer for their valuable questions and insights. We are pleased that they find our proposed method novel, resource-efficient, and highly generalizable.
>
> We have made several amendments and addressed the reviewer's specific queries as detailed below:
>
> __Q1__: "__Inference time cost / Slow speed.__""
>
> We appreciate the reviewer highlighting the concern regarding test-time computational costs. We acknowledge there is indeed a trade-off in test-time optimization speed and mask prompt quality. Specifically:
>
> 1. Previous methods typically require extensive resources and effort to train or fine-tune large vision-language models on training datasets. This approach is resource-intensive with limited generalization capabilities.
> 2. In contrast, our proposed test-time prompt optimization method is more efficient and better suited to real-world open-set applications, where test samples may contain novel linguistic and visual concepts. In such scenarios, textual embeddings need to be dynamically updated and aligned with the target object during testing. Consequently, this necessary trade-off between inference speed and embedding alignment cannot be entirely avoided.
> 3. However, we have demonstrated that it is possible to substantially reduce the number of test-time steps from 1100 to 50 by adapting the text encoder to the target dataset domain fine-tuned by only a samll number of samples. Furthermore, future work could focus on additional engineering optimizations, such as replacing the current backbone with HyperDiffusion, to further enhance inference speed.

---

### Decision · Program_Chairs · 2025-05-01

**Decision:**

Accept (poster)

**Comment:**

This paper introduces a training-free approach for open-set language-grounded image segmentation that leverages cross-attention maps from diffusion models and introduces language-guided visual prompt regularization to improve mask quality. Despite initial mixed scores (2, 2, 4, 4), the reviewers converged towards acceptance following the author response and discussion period. Concerns raised by the initially less positive reviewers were largely addressed, with one indicating an inclination to raise their score and the other expressing comfort with acceptance conditional on minor revisions, while the positive reviewers maintained their support after their concerns were resolved.

This paper presents an interesting direction and strong results on challenging benchmarks, making it a valuable contribution to ICML; the authors are requested to incorporate the remaining feedback from all reviewers into the final camera-ready version, paying particular attention to clarifying claims as suggested (especially for ixNi).